# Enhancing Preference-based Linear Bandits
# via Human Response Time

**Shen Li**[1]* **Yuyang Zhang**[2]* **Zhaolin Ren**[2] **Claire Liang**[1] **Na Li**[2] **Julie A. Shah**[1]

[1]Massachusetts Institute of Technology    [2]Harvard University

{shenli,cyl48}@mit.edu, julie_a_shah@csail.mit.edu
{yuyangzhang,zhaolinren}@g.harvard.edu, nali@seas.harvard.edu

## Abstract

Interactive preference learning systems infer human preferences by presenting queries as pairs of options and collecting binary choices. Although binary choices are simple and widely used, they provide limited information about preference strength. To address this, we leverage human response times, which are inversely related to preference strength, as an additional signal. We propose a computationally efficient method that combines choices and response times to estimate human utility functions, grounded in the EZ diffusion model from psychology. Theoretical and empirical analyses show that for queries with strong preferences, response times complement choices by providing extra information about preference strength, leading to significantly improved utility estimation. We incorporate this estimator into preference-based linear bandits for fixed-budget best-arm identification. Simulations on three real-world datasets demonstrate that using response times significantly accelerates preference learning compared to choice-only approaches. Additional materials, such as code, slides, and talk video, are available at `https://shenlirobot.github.io/pages/NeurIPS24.html`.

## 1 Introduction

Interactive preference learning from human binary choices is widely used in recommender systems [9, 21, 32, 56], assistive robots [54, 65], and fine-tuning large language models [5, 43, 46, 47, 59]. This process is often framed as a preference-based bandit problem [7, 31], where the system repeatedly presents queries as pairs of options, the human selects a preferred option, and the system infers preferences from these choices. Binary choices are popular because they are easy to implement and impose low cognitive load on users [37, 72, 74]. However, while binary choices reveal preferences, they provide little information about preference strength [77]. To address this, researchers have incorporated additional *explicit human feedback*, such as ratings [50, 58], labels [74], and slider bars [5, 72], but these approaches often complicate interfaces and increase cognitive demands [36, 37].

In this paper, we propose leveraging *implicit human feedback*, specifically response times, to provide additional insights into preference strength. Unlike explicit feedback, response time is unobtrusive and effortless to measure [17], offering valuable information that complements binary choices [2, 16]. For instance, consider an online retailer that repeatedly presents users with a binary query, whether to purchase or skip a recommended product [35]. Since most users skip products most of the time [33], the probability of skipping becomes nearly 1 for most items. This lack of variation in choices makes it difficult to assess how much a user likes or dislikes any specific product, limiting the system's ability to accurately infer their preferences. Response time can help overcome this limitation. Psychological research shows an inverse relationship between response time and preference strength [17]: users who strongly prefer to skip a product tend to do so quickly, while longer response times can indicate

---

*First two authors have equal contribution.

38th Conference on Neural Information Processing Systems (NeurIPS 2024).

weaker preferences. Thus, even when choices appear similar, response time can uncover subtle differences in preference strength, helping to accelerate preference learning.

Leveraging response times for preference learning presents notable challenges. Psychological research has extensively studied the relationship between human choices and response times [17, 19] using complex models like Drift-Diffusion Models [51] and Race Models [12, 66]. While these models align with both behavioral and neurobiological evidence [70], they rely on computationally intensive methods, such as hierarchical Bayesian inference [71] and maximum likelihood estimation (MLE) [52], to estimate the underlying human utility functions from both human choices and response times, making them impractical for real-time interactive systems. Although faster estimators exist [8, 28, 30, 67, 68], they typically estimate the utility functions for a single pair of options without aggregating data across multiple pairs. This limits their ability to leverage structures like linear utility functions, which are widely adopted both in preference learning with large option spaces [21, 24, 41, 54, 56] and in cognitive models for human multi-attribute decision-making [26, 64, 76].

To address these challenges, we propose a computationally efficient method for estimating linear human utility functions from both choices and response times, grounded in the difference-based EZ diffusion model [8, 67]. Our method leverages response times to transform binary choices into richer continuous signals, framing utility estimation as a *linear regression* problem that aggregates data across multiple pairs of options. We compare our estimator to traditional *logistic regression* methods that rely solely on choices [3, 31]. For queries with strong preferences, our theoretical and empirical analyses show that response times complement choices by providing additional information about preference strength. This significantly improves utility estimation compared to using choices alone. For queries with weak preferences, response times add little value but do not degrade performance. **In summary, response times complement choices, particularly for queries with strong preferences.**

Our linear-regression-based estimator integrates seamlessly into algorithms for preference-based bandits with linear human utility functions [3, 31], enabling interactive learning systems to leverage response times for faster learning. We specifically integrated our estimator into the Generalized Successive Elimination algorithm [3] for fixed-budget best-arm identification [29, 34]. Simulations using three real-world datasets [16, 39, 57] consistently show that incorporating response times significantly reduces identification errors, compared to traditional methods that rely solely on choices. *To the best of our knowledge, this is the first work to integrate response times into bandits (and RL).*

Section 2 introduces the preference-based linear bandit problem and the difference-based EZ diffusion model. Section 3 presents our utility estimator, incorporating both choices and response times, and offers a theoretical comparison to the choice-only estimator. Section 4 integrates both estimators into the Generalized Successive Elimination algorithm. Section 5 presents empirical results for estimation and bandit learning. Section 6 discusses the limitations of our approach. Appendix B reviews response time models, parameter estimation techniques, and their connection to preference-based RL.

*Nomenclature*: We use $[n]$ to denote the set $\{1, \ldots, n\}$. For a scalar random variable $x$, the expectation and variance are denoted by $\mathbb{E}[x]$ and $\mathbb{V}[x]$, respectively. The function $\mathrm{sgn}(x)$ denotes the sign of $x$.

## 2 Problem setting and preliminaries

**Preference-based bandits with a linear utility function.** The learner is given a finite set of options (or "arms"), each represented by a feature vector in $\mathcal{Z} \subset \mathbb{R}^d$, and a finite set of binary queries, where each query is the difference between two arms, denoted by $\mathcal{X} \subset \mathbb{R}^d$. For instance, if the learner can query any pair of arms, the query space is $\mathcal{X} = \{z - z' : z, z' \in \mathcal{Z}\}$. In the online retailer example from section 1, the query space is $\mathcal{X} = \{z - z_{\mathrm{skip}} : z \in \mathcal{Z}\}$, where $z$ represents purchasing a product and $z_{\mathrm{skip}}$ represents skipping (often set as $\mathbf{0}$). For each arm $z \in \mathcal{Z}$, the human utility is assumed to be linear in the feature space, defined as $u_z := z^\top \theta^*$, where $\theta^* \in \mathbb{R}^d$ represents the human's preference parameters. For any query $x \in \mathcal{X}$, the utility difference is then defined as $u_x := x^\top \theta^*$.

Given a query $x := z_1 - z_2 \in \mathcal{X}$, we model human choices and response times using the difference-based EZ-Diffusion Model (dEZDM) [8, 67], integrated with our linear utility structure. (See appendix B.1 for a comparison with other models.) This model interprets human decision-making as a stochastic process in which evidence accumulates over time to compare two options. As shown in fig. 1a, after receiving a query $x$, the human first spends a fixed amount of non-decision time, denoted by $t_{\mathrm{nondec}} > 0$, to perceive and encode the query. Then, evidence $E_x$ accumulates over

time following a Brownian motion with drift $x^\top\theta^*$ and two symmetric absorbing barriers, $a > 0$ and $-a$. Specifically, at time $t_{\text{nondec}} + \tau$ where $\tau \geq 0$, the evidence is $E_{x,\tau} = x^\top\theta^* \cdot \tau + B(\tau)$, where $B(\tau) \sim \mathcal{N}(0, \tau)$ is standard Brownian motion. This process continues until the evidence reaches either the upper barrier $a$ or lower barrier $-a$, at which point a decision is made. The random stopping time, $t_x := \min\{\tau > 0 \colon E_{x,\tau} \in \{a, -a\}\}$, represents the decision time. If $E_{x,t_x} = a$, the human chooses $z_1$; if $E_{x,t_x} = -a$, they choose $z_2$. The choice is represented by the random variable $c_x$, where $c_x = 1$ if $z_1$ is chosen, and $-1$ if $z_2$ is chosen. The total response time, $t_{\text{RT},x}$, is the sum of the non-decision time and the decision time: $t_{\text{RT},x} = t_{\text{nondec}} + t_x$. The choice probability, expected choice, choice variance, and expected decision time are given as follows [48, eq. (A.16) and (A.17)]:

$$\forall x \in \mathcal{X} \colon \mathbb{P}\left[c_x = 1\right] = \frac{1}{1 + \exp(-2ax^\top\theta^*)}, \quad \mathbb{E}\left[c_x\right] = \tanh(ax^\top\theta^*)$$

$$\mathbb{V}\left[c_x\right] = 1 - \tanh^2(ax^\top\theta^*), \quad \mathbb{E}\left[t_x\right] = \begin{cases} \frac{a}{x^\top\theta^*}\tanh(ax^\top\theta^*) & \text{if } x^\top\theta^* \neq 0 \\ a^2 & \text{if } x^\top\theta^* = 0 \end{cases}. \tag{1}$$

This choice probability matches that of the Bradley and Terry [10] model. If the learner relies solely on choices, then our bandit problem reduces to the transductive linear logistic bandit problem [31].

Figures 1b and 1c illustrate the roles of the parameters $x^\top\theta^*$ and $a$. First, the absolute drift (or the absolute utility difference), $|x^\top\theta^*|$, reflects the human's preference strength for the query $x$. Larger values indicate stronger preferences, leading to faster decisions and more consistent choices. Smaller values suggest weaker preferences, resulting in slower decisions and less consistent choices. Second, the barrier $a$ represents the human's conservativeness in decision-making [40]. A more conservative human (higher $a$) requires more evidence to decide, resulting in slower but more consistent choices. In contrast, a less conservative human (lower $a$) decides faster but makes less consistent choices.

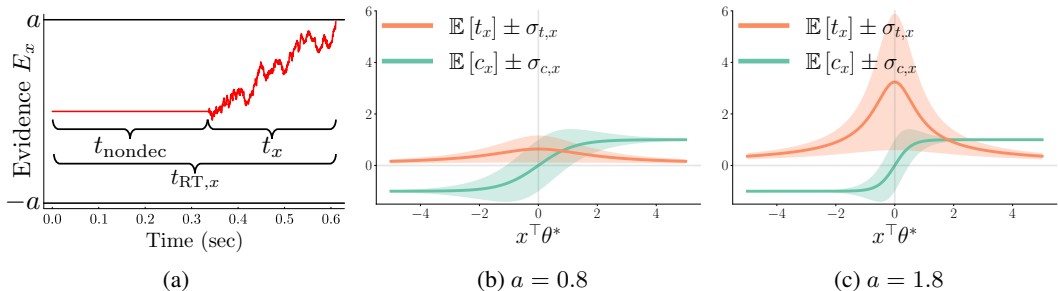

(a)  (b) $a = 0.8$  (c) $a = 1.8$

Figure 1: (a) depicts the human decision-making process for a binary query $x \in \mathcal{X}$, where the human selects between two arms. The human first spends a fixed non-decision time $t_{\text{nondec}}$ encoding the query. Then, the human's evidence accumulates according to a Brownian motion with drift $x^\top\theta^*$. When the evidence reaches the upper barrier $a$ or lower barrier $-a$, the human makes a choice, denoted by $c_x = 1$ or $c_x = -1$, respectively. The random stopping time of the accumulation process is the decision time $t_x$, and the total response time is $t_{\text{RT},x} = t_{\text{nondec}} + t_x$. (b) and (c) plot the expected choice $\mathbb{E}[c_x]$ and the expected decision time $\mathbb{E}[t_x]$, with shaded regions representing one standard deviation, plotted as functions of the utility difference $x^\top\theta^*$ for two barrier values $a$.

We adopt the common assumption that $t_{\text{nondec}}$ is constant across all queries for a given human [16, 76] and further assume that $t_{\text{nondec}}$ is known to the learner. This assumption enables the learner to perfectly recover $t_x$ from the observed $t_{\text{RT},x}$. In section 5.2, we empirically show that even when $t_{\text{nondec}}$ is unknown, its impact on the performance of our method that relies on decision times is negligible.

**Learning objective: Best-arm identification with a fixed budget.** We focus on the fixed-budget best-arm identification problem [29, 34]. The learner is provided with a total interaction time budget $B > 0$, an arm space $\mathcal{Z}$, a query space $\mathcal{X}$, and a non-decision time $t_{\text{nondec}}$. Both the human's preference vector $\theta^*$ and the decision barrier $a$ are unknown. In each episode $s \in \mathbb{N}$, the learner selects a query $x_s \in \mathcal{X}$, receives human feedback $(c_{x_s,s}, t_{x_s,s})$ generated by the dEZDM, and consumes $t_{\text{RT},x_s,s}$ time. When the cumulative interaction time exceeds the budget $B$ at some episode $S$, i.e., $\sum_{s=1}^{S} t_{\text{RT},x_s,s} > B$, the learner must stop and recommend an arm $\hat{z} \in \mathcal{Z}$. The goal is to recommend the unique best arm $z^* := \arg\max_{z \in \mathcal{Z}} z^\top\theta^*$, minimizing the error probability $\mathbb{P}\left[\hat{z} \neq z^*\right]$.

To address this problem, we adopt the Generalized Successive Elimination (GSE) algorithm [1, 3, 75]. GSE divides the total budget $B$ into multiple phases. In each phase, it strategically samples queries until the phase's budget is exhausted, collecting both human choices and decision times. It then estimates the preference vector $\theta^*$ and eliminates arms with low estimated utilities. Decision times play a key role in the estimation step by providing complementary information about preference strength, which can enable more accurate estimation of $\theta^*$ than choices alone. Next, in section 3, we introduce a novel estimator that combines decision times and choices to estimate $\theta^*$. Then, in section 4, we discuss how this estimator is integrated into GSE to improve preference learning.

## 3 Utility estimation

This section addresses the problem of estimating human preference $\theta^*$ from a fixed dataset, denoted by $\left\{x, c_{x,s_{x,i}}, t_{x,s_{x,i}}\right\}_{x \in \mathcal{X}_{\text{sample}}, i \in [n_x]}$. Here, $\mathcal{X}_{\text{sample}}$ denotes the set of queries in the dataset, $n_x$ denotes the number of samples for each query $x \in \mathcal{X}_{\text{sample}}$, and $s_{x,i}$ denotes the episode when $x$ is sampled for the $i$-th time. Samples from the same query $x$ are i.i.d., while samples from different queries are independent. Section 3.1 introduces a new estimator, the "choice-decision-time estimator," which uses both choices and decision times, in contrast to the commonly used "choice-only estimator" that only uses choices [3, 31]. Sections 3.2 and 3.3 theoretically compares these estimators, analyzing both asymptotic and non-asymptotic performance and highlighting the advantages of incorporating decision times. Section 5.1 presents empirical results that validate our theoretical insights.

### 3.1 Choice-decision-time estimator and choice-only estimator

The choice-decision-time estimator is based on the following relationship between human utilities, choices, and decision times, derived from eq. (1):

$$\forall x \in \mathcal{X}: x^\top \frac{\theta^*}{a} = \frac{\mathbb{E}\left[c_x\right]}{\mathbb{E}\left[t_x\right]}. \tag{2}$$

Intuitively, when a human provides consistent choices (i.e., large $|\mathbb{E}[c_x]|$) and makes decisions quickly (i.e., small $\mathbb{E}[t_x]$), it implies a strong preference (i.e., large $|x^\top \theta^*|$). This relationship formulates the estimation of $\theta^*$ as a *linear regression* problem. Accordingly, the choice-decision-time estimator calculates the empirical means of both choices and decision times, aggregates the ratios across all sampled queries, and applies ordinary least squares (OLS) to estimate $\theta^*/a$. Since the ranking of arm utilities based on $\theta^*/a$ is identical to that based on $\theta^*$, estimating $\theta^*/a$ is sufficient for identifying the best arm. Formally, this estimate of $\theta^*/a$, denoted by $\widehat{\theta}_{\text{CH,DT}}$, is given by:

$$\widehat{\theta}_{\text{CH,DT}} := \left(\sum_{x \in \mathcal{X}_{\text{sample}}} n_x\, xx^\top\right)^{-1} \sum_{x \in \mathcal{X}_{\text{sample}}} n_x\, x\, \frac{\sum_{i=1}^{n_x} c_{x,s_{x,i}}}{\sum_{i=1}^{n_x} t_{x,s_{x,i}}}. \tag{3}$$

In contrast, the choice-only estimator is based on eq. (1), which shows that for each query $x \in \mathcal{X}$, the random variable $(c_x + 1)/2$ follows a Bernoulli distribution with mean $1/[1 + \exp(-x^\top \cdot 2a\theta^*)]$. Similar to the choice-decision-time estimator, the parameter $2a$ does not impact the ranking of arms, so estimating $2a\theta^*$ is sufficient for best-arm identification. This estimation is formulated as a *logistic regression* problem [3, 31], with MLE providing the following estimate of $2a\theta^*$, denoted by $\widehat{\theta}_{\text{CH}}$:

$$\widehat{\theta}_{\text{CH}} := \arg\max_{\theta \in \mathbb{R}^d} \sum_{x \in \mathcal{X}_{\text{sample}}} \sum_{i=1}^{n_x} \log \mu(c_{x,s_{x,i}}\, x^\top \theta), \tag{4}$$

where $\mu(y) := 1/[1 + \exp(-y)]$ is the standard logistic function. While this MLE lacks a closed-form solution, it can be efficiently solved using optimization methods like Newton's algorithm [25, 44].

### 3.2 Asymptotic normality of the two estimators

The choice-decision-time estimator from eq. (3) satisfies the following asymptotic normality result:

**Theorem 3.1** (Asymptotic normality of $\widehat{\theta}_{\text{CH,DT}}$)**.** *Given a fixed i.i.d. dataset $\left\{x, c_{x,s_{x,i}}, t_{x,s_{x,i}}\right\}_{i\in[n]}$ for each $x \in \mathcal{X}_{sample}$, where $\sum_{x\in\mathcal{X}_{sample}} xx^\top \succ 0$, and assuming that the datasets for different $x \in \mathcal{X}_{sample}$ are independent, then, for any vector $y \in \mathbb{R}^d$, as $n \to \infty$, the following holds:*

$$\sqrt{n}\, y^\top \left(\widehat{\theta}_{CH,DT,n} - \theta^*/a\right) \xrightarrow{D} \mathcal{N}(0, \zeta^2/a^2).$$

*Here, the asymptotic variance depends on a problem-specific constant, $\zeta^2$, with an upper bounded:*

$$\zeta^2 \leq \|y\|^2_{\left(\sum_{x\in\mathcal{X}_{sample}}\left[\min_{x'\in\mathcal{X}_{sample}} \mathbb{E}[t_{x'}]\right]\cdot xx^\top\right)^{-1}}.$$

The proof is provided in appendix C.2. The asymptotic variance upper bound shows that all sampled queries are weighted by a common factor $\min_{x'\in\mathcal{X}_{sample}} \mathbb{E}[t_{x'}]$, which is the smallest expected decision time among all the sampled queries in $\mathcal{X}_{sample}$. This weight represents the amount of information provided by each query's choices and decision times for utility estimation. A larger weight indicates that all queries in $\mathcal{X}_{sample}$ provides more information, leading to lower variance and better estimates.

In contrast, the choice-only estimator from eq. (4) has the following asymptotic normality result, as derived from Fahrmeir and Kaufmann [23, corollary 1]:

**Theorem 3.2** (Asymptotic normality of $\widehat{\theta}_{\text{CH}}$)**.** *Given a fixed i.i.d. dataset $\left\{x, c_{x,s_{x,i}}, t_{x,s_{x,i}}\right\}_{i\in[n]}$ for each $x \in \mathcal{X}_{sample}$, where $\sum_{x\in\mathcal{X}_{sample}} xx^\top \succ 0$, and assuming that the datasets for different $x \in \mathcal{X}_{sample}$ are independent, then, for any vector $y \in \mathbb{R}^d$, as $n \to \infty$, the following holds:*

$$\sqrt{n}y^\top \left(\widehat{\theta}_{CH,n} - 2a\theta^*\right) \xrightarrow{D} \mathcal{N}\left(0, 4a^2\,\|y\|^2_{\left(\sum_{x\in\mathcal{X}_{sample}}[a^2\,\mathbb{V}[c_x]]\cdot xx^\top\right)^{-1}}\right).$$

This asymptotic variance shows that each sampled query $x \in \mathcal{X}_{sample}$ is weighted by its own factor $a^2\,\mathbb{V}[c_x]$, representing the amount of information the query's choices contribute to utility estimation. A larger weight indicates that the query contributes more information, leading to better estimates.

The weights in both theorems highlight the different contributions of choices and decision times to utility estimation. In the choice-only estimator (theorem 3.2), each query is weighted by $a^2\,\mathbb{V}[c_x]$, which depends on the utility difference $x^\top\theta^*$ for a fixed barrier $a$. As shown by the gray curves in fig. 2a, this weight quickly decays to zero as preferences become stronger (i.e., as $|x^\top\theta^*|$ increases). This indicates that *choices from queries with strong preferences provide little information.* Intuitively, when preferences are strong, humans consistently select the same option, making it hard to distinguish whether their preference is moderately or very strong. As a result, choices from such queries contribute minimally to utility estimation. This intuition aligns with the online retailer example in section 1.

For the choice-decision-time estimator (theorem 3.1), queries are weighted by $\min_{x'\in\mathcal{X}_{sample}} \mathbb{E}[t_{x'}]$, which depends on both $\mathcal{X}_{sample}$ and $\mathbb{E}[t_x]$. To better understand this weight, we first plot $\mathbb{E}[t_x]$ without the 'min' operator as the orange curves in fig. 2a. Comparing the orange and gray curves shows that $\mathbb{E}[t_x]$ is generally larger than the choice-only weight, $a^2\,\mathbb{V}[c_x]$. The actual weight in the choice-decision-time estimator, which is the minimum expected decision time across sampled queries, is less than or equal to the orange curve but is likely still higher than the choice-only weight, especially for queries with strong preferences. This suggests that *when preferences are strong, decision times complement choices by capturing preference strength, leading to improved estimation.*

When queries have weak preferences, the choice-decision-time weight may be lower than the choice-only weight. However, since the choice-decision-time weight represents only an upper bound on the asymptotic variance (theorem 3.1), no definitive conclusions can be drawn from the theory alone. Empirically, as shown in section 5.1, decision times add little value but do not degrade performance.

As the barrier $a$ increases, the choice-decision-time weight rises. In contrast, the choice-only weight increases for queries with weak preferences, but this increase is concentrated in a narrower region, with weights decreasing elsewhere. Intuitively, a higher barrier reflects greater conservativeness in human decision-making, leading to longer decision times and more consistent choices (fig. 1). As a result, more queries exhibit strong preferences, making choices from these queries less informative.

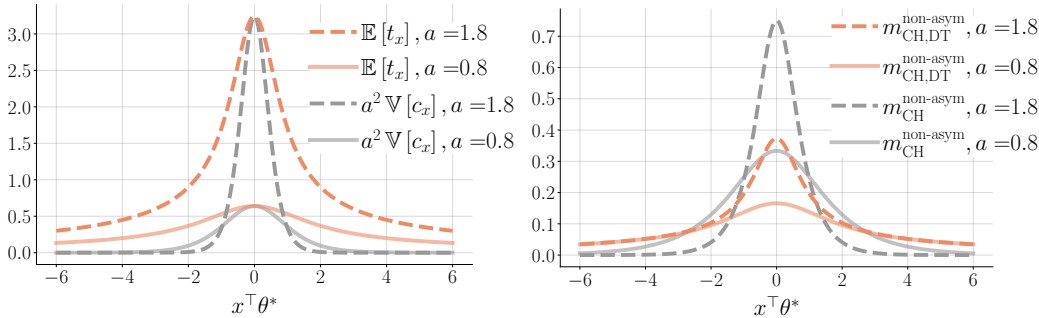

(a) $\mathbb{E}[t_x]$ and $a^2 \mathbb{V}[c_x]$ in asymptotic variances

(b) Weights in non-asymptotic concentration bounds

Figure 2: This figure illustrates key terms from our theoretical analyses, highlighting the different contributions of choices and decision times to utility estimation. These terms are functions of the utility difference $x^\top \theta^*$ and are plotted for two barrier values, $a$. (a) compares the weights $\mathbb{E}[t_x]$ and $a^2 \mathbb{V}[c_x]$ in the asymptotic variances for the choice-decision-time estimator (orange, theorem 3.1) and the choice-only estimator (gray, theorem 3.2), respectively. This comparison shows that *decision times complement choices, particularly for queries with strong preferences*. (b) compares the weights in the non-asymptotic concentration bounds (theorems 3.3 and 3.4), showing similar trends, though these weights may not be optimal due to proof techniques.

## 3.3 Non-asymptotic concentration of the two estimators for utility difference estimation

In this section, we focus on the simpler problem of estimating the utility difference for a single query, without aggregating data from multiple queries. Comparing the non-asymptotic concentration bounds of both estimators, in this case, provides insights similar to those discussed in section 3.2. Extending this non-asymptotic analysis to the full estimation of the preference vector $\theta^*$ is left for future work.

Given a query $x \in \mathcal{X}$, the task is to estimate the utility difference $u_x := x^\top \theta^*$ using the fixed i.i.d. dataset $\{(c_{x,s_{x,i}}, t_{x,s_{x,i}})\}_{i \in [n_x]}$. Applying the choice-decision-time estimator from eq. (3), we get the following estimate (for details, see appendix C.3.1), which estimates $u_x/a$ rather than $u_x$:

$$\widehat{u}_{x,\text{CH,DT}} := \frac{\sum_{i=1}^{n_x} c_{x,s_{x,i}}}{\sum_{i=1}^{n_x} t_{x,s_{x,i}}}. \tag{5}$$

In contrast, applying the choice-only estimator from eq. (4), we get the following estimate (for details, see appendix C.3.2), which estimates $2au_x$ rather than $u_x$:

$$\widehat{u}_{x,\text{CH}} := \mu^{-1} \left( \frac{1}{n_x} \sum_{i=1}^{n_x} \frac{c_{x,s_{x,i}} + 1}{2} \right), \tag{6}$$

where $(c_{x,s_{x,i}} + 1)/2$ is the binary choice coded as 0 or 1, and $\mu^{-1}(p) := \log(p/(1-p))$ is the logit function (inverse of $\mu$ introduced in eq. (4)).

Notably, the choice-only estimator in eq. (6) aligns with the EZ-diffusion model's drift estimator [67, eq. (5)]. Moreover, the estimators in Xiang Chiong et al. [73, eq. (6)] and Berlinghieri et al. [8, eq. (7)] combine elements of both estimators from eqs. (5) and (6). In section 5.2, we demonstrate that both estimators from Wagenmakers et al. [67, eq. (5)] and Xiang Chiong et al. [73, eq. (6)] are outperformed by our proposed estimator in eq. (3) for the full bandit problem.

Assuming the utility difference $u_x \neq 0$, the choice-decision-time estimator in eq. (5) satisfies the following non-asymptotic concentration bound, proven in appendix C.3.1:

**Theorem 3.3** (Non-asymptotic concentration of $\widehat{u}_{x,\text{CH,DT}}$). *For each query $x \in \mathcal{X}$ with $u_x \neq 0$, given a fixed i.i.d. dataset $\{(c_{x,s_{x,i}}, t_{x,s_{x,i}})\}_{i \in [n_x]}$, for any $\epsilon > 0$ satisfying $\epsilon \leq \min\{|u_x|/(\sqrt{2}a), (1+\sqrt{2})a|u_x|/\mathbb{E}[t_x]\}$, the following holds:*

$$\mathbb{P}\left( \left| \widehat{u}_{x,CH,DT} - \frac{u_x}{a} \right| > \epsilon \right) \leq 4 \exp\left( - \left[ m_{CH,DT}^{non\text{-}asym}(x^\top \theta^*) \right]^2 n_x \left[ \epsilon \cdot a \right]^2 \right),$$

*where $m_{CH,DT}^{non\text{-}asym}(x^\top \theta^*) := \mathbb{E}[t_x] / [(2 + 2\sqrt{2})a]$.*

In contrast, the choice-only estimator in eq. (6) has the following non-asymptotic concentration result, adapted from Jun et al. [31, theorem 5][2]:

**Theorem 3.4** (Non-asymptotic concentration of $\widehat{u}_{x,\mathrm{CH}}$)**.** *For each query* $x \in \mathcal{X}$, *given a fixed i.i.d. dataset* $\left\{c_{x,s_{x,i}}\right\}_{i \in [n_x]}$, *for any positive* $\epsilon < 0.3$, *if* $n_x \geq 1/\dot{\mu}(2au_x) \cdot \max\{3^2 \log(6e)/\epsilon^2, 64 \log(3)/(1 - \epsilon^2/0.3^2)\}$, *the following holds:*

$$\mathbb{P}\left(|\widehat{u}_{x,CH} - 2au_x| > \epsilon\right) \leq 6 \exp\left(-\left[m_{CH}^{non\text{-}asym}\left(x^\top \theta^*\right)\right]^2 n_x \left[\epsilon/(2a)\right]^2\right),$$

*where* $m_{CH}^{non\text{-}asym}\left(x^\top \theta^*\right) := a \sqrt{\mathbb{V}\left[c_x\right]}/2.4$.

The weights $m_{\mathrm{CH,DT}}^{\text{non-asym}}(\cdot)$ and $m_{\mathrm{CH}}^{\text{non-asym}}(\cdot)$ from theorems 3.3 and 3.4, respectively, are functions of the utility difference $x^\top \theta^*$ for a fixed barrier $a$. These weights determine how quickly estimation errors decay as the dataset size $n_x$ grows, with larger weights indicating faster error reduction. While these weights may not be optimal due to proof techniques, they highlight the distinct contributions of choices and decision times, consistent with our asymptotic analysis in section 3.2. Figure 2b compares the weights for the choice-decision-time estimator (orange, $m_{\mathrm{CH,DT}}^{\text{non-asym}}(\cdot)$) and the choice-only estimator (gray, $m_{\mathrm{CH}}^{\text{non-asym}}(\cdot)$). For strong preferences, the choice-only weights quickly decay to zero, while the choice-decision-time weights remain relatively large. This supports our key insight that decision times complement choices and improve estimation for queries with strong preferences.

In summary, both asymptotic (section 3.2) and non-asymptotic (section 3.3) analyses demonstrate that the choice-decision-time estimator extracts more information from queries with strong preferences. This finding aligns with prior empirical work [16] and is further supported by our results in section 5.1.

In fixed-budget best-arm identification, our choice-decision-time estimator's ability to extract more information from queries with strong preferences is especially valuable. Bandit learners, such as GSE [3], strategically sample queries, update estimates of $\theta^*$, and eliminate lower-utility arms. With the choice-only estimator, learners struggle to extract information from queries with strong preferences. To resolve this, one approach is to selectively sample queries with weak preferences, but this has two drawbacks. First, queries with weak preferences take longer to answer (i.e., require more resources), potentially lowering the 'bang per buck' (information per resource) [4]. Second, since $\theta^*$ is unknown in advance, learners cannot reliably target queries with weak preferences. In contrast, with our choice-decision-time estimator, learners leverage decision times to gain more information from queries with strong preferences, improving bandit learning performance. We integrate both estimators into bandit learning in section 4 and evaluate their performance in section 5.

## 4 Interactive learning algorithm

We introduce the Generalized Successive Elimination (GSE) algorithm [1, 3, 75] for fixed-budget best-arm identification in preference-based linear bandits, and outline the key options for each GSE component, which we empirically compare in section 5.

The pseudo-code for GSE is shown in algorithm 1. The algorithm uses a hyperparameter $\eta$ to control the number of phases, the budget per phase, and the number of arms eliminated in each phase. GSE divides the total budget $B$ evenly across phases and reserves a buffer, sized by another hyperparameter $B_{\mathrm{buff}}$, to prevent overspending in any phase (line 4). In each phase, GSE computes an experimental design $\lambda$, a probability distribution over the query space, to guide query sampling. We consider two designs: the transductive design [24], $\lambda_{\mathrm{trans}}$ (line 5), and the weak-preference design [31], $\lambda_{\mathrm{weak}}$ (line 6). Both designs minimize the worst-case variance of utility differences between surviving arms. The transductive design weights all queries equally, whereas the weak-preference design prioritizes queries with weak preferences to counter the choice-only estimator's difficulty in extracting information from queries with strong preferences (section 3). Since $\theta^*$ is unknown, the weak-preference design identifies queries with weak preferences based on the previous phase's estimate $\widehat{\theta}_{\mathrm{CH}}$. Then, GSE samples queries based on the design (line 7) and, after exhausting the phase's budget, estimates $\theta^*$ using either the choice-decision-time estimator $\widehat{\theta}_{\mathrm{CH,DT}}$ (line 8) or the choice-only estimator $\widehat{\theta}_{\mathrm{CH}}$ (line 9). It then eliminates arms with low estimated utilities (line 10). This process repeats until only one arm remains, which GSE recommends as the best arm (line 12).

---

[2] In Jun et al. [31, theorem 5], we let $x_1 = \cdots = x_t = 1$ and $t_{\mathrm{eff}} = d = 1$.

The key difference between algorithm 1 and previous GSE algorithms [1, 3, 75] is that our setting involves queries with random response times, unknown to the learner. Previous work assumes fixed resource consumption per query and uses deterministic rounding methods [3, 24] to pre-allocate queries. This approach does not handle random resource usage. Instead, we adopt a random sampling procedure [13, 61] in line 7 to allocate queries based on the design. Random resource usage also requires tuning the elimination parameter $\eta$, to balance data collection and arm elimination, and the buffer size $B_{\text{buff}}$, to prevent overspending. In our empirical study (section 5.2), we manually tune both parameters. Further theoretical analysis is needed to better understand and optimize them.

---

**Algorithm 1** Generalized Successive Elimination (GSE) [3]

---

1: **Input:** Arm space $\mathcal{Z}$, query space $\mathcal{X}$, non-decision time $t_{\text{nondec}}$, and total budget $B$.
2: **Hyperparameters:** Elimination parameter $\eta$ and buffer size $B_{\text{buff}}$.
3: **Initialization:** $\mathcal{Z}_1 \leftarrow \mathcal{Z}$.
4: **for** each phase $k = 1, \ldots, K := \lceil \log_\eta |\mathcal{Z}| \rceil$ with the budget $B_k := B/K - B_{\text{buff}}$ **do**
5:    Design 1. $\lambda_k := \lambda_{\text{trans},k} \leftarrow \arg\min_{\lambda \in \blacktriangle^{|\mathcal{X}|}} \max_{z \neq z' \in \mathcal{Z}_k} \|z - z'\|^2_{(\sum_{x \in \mathcal{X}} \lambda_x xx^\top)^{-1}}$.
6:    Design 2. $\lambda_k := \lambda_{\text{weak},k} \leftarrow \arg\min_{\lambda \in \blacktriangle^{|\mathcal{X}|}} \max_{z \neq z' \in \mathcal{Z}_k} \|z - z'\|^2_{(\sum_{x \in \mathcal{X}} \dot{\mu}(x^\top \widehat{\theta}_{k-1}) \lambda_x xx^\top)^{-1}}$.
7:    Sample queries $x_j \sim \lambda_k$ and stop at $J_k$ if $\sum_{j=1}^{J_k-1} t_{\text{RT},x_j,j} \leq B_k$ and $\sum_{j=1}^{J_k} t_{\text{RT},x_j,j} > B_k$.
8:    Estimate 1. $\widehat{\theta}_k := \widehat{\theta}_{\text{CH,DT},k} \leftarrow$ apply eq. (3) to all the $J_k$ samples.
9:    Estimate 2. $\widehat{\theta}_k := \widehat{\theta}_{\text{CH},k} \leftarrow$ apply eq. (4) to all the $J_k$ samples.
10:    Update $\mathcal{Z}_{k+1} \leftarrow$ Top-$\lceil \frac{|\mathcal{Z}_k|}{\eta} \rceil$ arms in $\mathcal{Z}_k$, ranked by the estimated utility $z^\top \widehat{\theta}_k$.
11: **end for**
12: **Output:** the single one $\widehat{z} \in \mathcal{Z}_{K+1}$.

---

# 5 Empirical results

This section empirically compares the GSE variations introduced in section 4: (1) $(\lambda_{\text{trans}}, \widehat{\theta}_{\text{CH,DT}})$: Transductive design with choice-decision-time estimator. (2) $(\lambda_{\text{trans}}, \widehat{\theta}_{\text{CH}})$: Transductive design with choice-only estimator. (3) $(\lambda_{\text{weak}}, \widehat{\theta}_{\text{CH}})$: Weak-preference design with choice-only estimator.

## 5.1 Estimation performance on synthetic data

We evaluate the estimation performance of the GSE variations on the "sphere" synthetic problem, a standard linear bandit problem in the literature [20, 42, 61]. Details are provided in appendix D.1.

Estimation performance, as discussed in section 3, depends on the utility difference $x^\top \theta^*$ and the barrier $a$. We vary $a$ over a range of values commonly used in psychology [16, 71]. To examine how preference strength impacts estimation, we scale each arm $z$ to $c_{\mathcal{Z}} \cdot z$, effectively scaling each utility difference $x^\top \theta^*$ to $c_{\mathcal{Z}} \cdot x^\top \theta^*$. Small $c_{\mathcal{Z}}$ values correspond to problems with weak preferences, while large values correspond to strong preferences. For each $(c_{\mathcal{Z}}, a)$ pair, the system generates 100 random problem instances and runs 100 repeated simulations per instance. In each simulation, the GSE variations sample 50 queries, ignoring the response time budget, and compute $\widehat{\theta}$. Performance is evaluated by $\mathbb{P}[\arg\max_{z \in \mathcal{Z}} z^\top \widehat{\theta} \neq z^*]$, which reflects the best-arm identification goal defined in section 2. To isolate the effect of estimation, we allow $\lambda_{\text{weak}}$ access to the true $\theta^*$, enabling it to perfectly compute the terms $\dot{\mu}(x^\top \theta^*)$ used in line 6 of algorithm 1.

As shown in fig. 3a, fixing the barrier $a$ and examining the vertical line, as $c_{\mathcal{Z}}$ increases and preferences become stronger, the performance of the choice-only estimator with the transductive design first improves and then declines. The initial improvement arises because larger $c_{\mathcal{Z}}$ increases utility differences between the best arm and others, theoretically simplifying best-arm identification. The subsequent decline, highlighted by the dark curved band, supports our insight from section 3 that choices from queries with strong preferences provide limited information. Fixing $c_{\mathcal{Z}}$ and examining the horizontal line, performance first improves and then declines. This trend aligns with fig. 2a and section 3.2, where higher barriers $a$ increase the choice-only weights for queries with weak

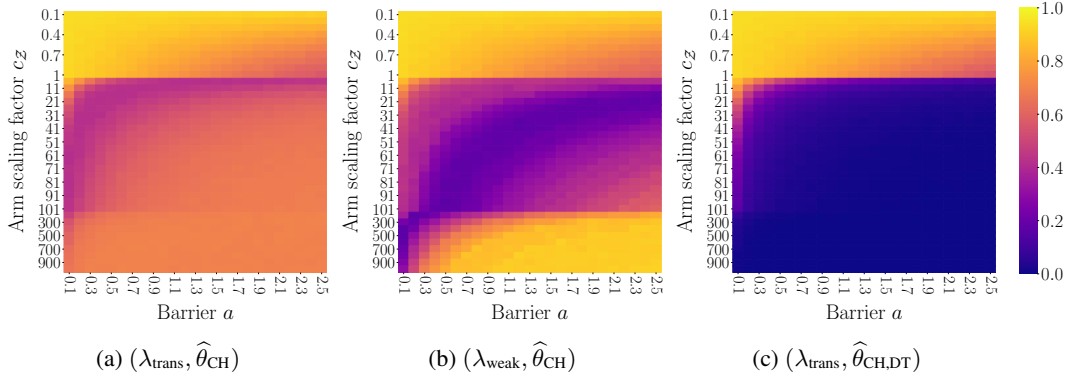

Figure 3: Three heatmaps show estimation error probabilities, $\mathbb{P}[\arg\max_{z\in\mathcal{Z}} z^\top\widehat{\theta} \neq z^*]$, for three GSE variations, shown as functions of the arm scaling factor $c_\mathcal{Z}$ and barrier $a$. Darker colors indicate better estimation. (a) The choice-only estimator $\widehat{\theta}_{\mathrm{CH}}$ with the transductive design $\lambda_{\mathrm{trans}}$ struggles as $c_\mathcal{Z}$ increases (i.e., preferences become stronger), highlighting that choices from queries with strong preferences provide limited information. (b) The weak-preference design $\lambda_{\mathrm{weak}}$ improves (a) by sampling queries with weak preferences but assumes perfect knowledge of $\theta^*$ and equal resource consumption across queries. (c) The choice-decision-time estimator $\widehat{\theta}_{\mathrm{CH,DT}}$ with $\lambda_{\mathrm{trans}}$ outperforms both choice-only methods in (a) and (b), showing that decision times complement choices and improve estimation, especially for strong preferences.

preferences, initially improving performance. However, as $a$ grows, fewer queries exhibit increased weights, while most queries' weights decrease, leading to the later performance drop.

In Figure 3b, for moderate $c_\mathcal{Z}$, the choice-only estimator with the weak-preference design outperforms the transductive design (fig. 3a), demonstrating that focusing on queries with weak preferences improves estimation. However, as $c_\mathcal{Z}$ becomes too large, performance declines because many $\dot{\mu}(x^\top\theta^*)$ in line 6 of algorithm 1 approach zero, preventing informative queries from being sampled. This advantage of the weak-preference design assumes perfect knowledge of $\theta^*$ and equal resource consumption across queries. In practice, where $\theta^*$ is unknown and weak-preference queries require longer response times, the transductive design performs better, as shown in section 5.2.

Figure 3c shows that the choice-decision-time estimator consistently outperforms the choice-only estimators under both the transductive and weak-preference designs, particularly for strong preferences. This suggests that for queries with strong preferences, decision times complement choices and improve estimation, confirming our theoretical insights from section 3, while for queries with weak preferences, decision times add little value but do not degrade performance. The performance also improves with a higher barrier $a$, supporting the insights conveyed by fig. 2a and section 3.2.

## 5.2 Fixed-budget best-arm identification performance on real datasets

This section compares the bandit performance of six GSE variations. The first three are as previously defined at the beginning of section 5: $(\lambda_{\mathrm{trans}}, \widehat{\theta}_{\mathrm{CH,DT}})$, $(\lambda_{\mathrm{trans}}, \widehat{\theta}_{\mathrm{CH}})$, and $(\lambda_{\mathrm{weak}}, \widehat{\theta}_{\mathrm{CH}})$.

The 4th GSE variation, $(\lambda_{\mathrm{trans}}, \widehat{\theta}_{\mathrm{CH,\mathbb{RT}}})$, evaluates the performance of the choice-decision-time estimator when the non-decision time $t_{\mathrm{nondec}}$ is unknown. The estimator, $\widehat{\theta}_{\mathrm{CH,\mathbb{RT}}}$, is identical to the original choice-decision-time estimator from Eq. (3), but with response times used in place of decision times.

The 5th GSE variation, $(\lambda_{\mathrm{trans}}, \widehat{\theta}_{\mathrm{CH,logit}})$, is based on Wagenmakers et al. [67, eq. (5)], which states that $x^\top \cdot (2a\theta^*) = \mu^{-1}(\mathbb{P}[c_x = 1])$, where $\mu^{-1}(p) := \log(p/(1-p))$. By incorporating our linear utility structure, we obtain the following choice-only estimator $\widehat{\theta}_{\mathrm{CH,logit}}$:

$$\widehat{\theta}_{\mathrm{CH,logit}} := \left( \sum_{x\in\mathcal{X}_{\mathrm{sample}}} n_x\, xx^\top \right)^{-1} \sum_{x\in\mathcal{X}_{\mathrm{sample}}} n_x\, x \cdot \mu^{-1}\left(\widehat{\mathfrak{c}}_x\right),$$

where $\widehat{\mathfrak{c}}_x := \frac{1}{n_x}\sum_{i=1}^{n_x} \frac{1}{2}\left(c_{x,s_{x,i}} + 1\right)$ is the empirical mean of the binary choices coded as 0 or 1.

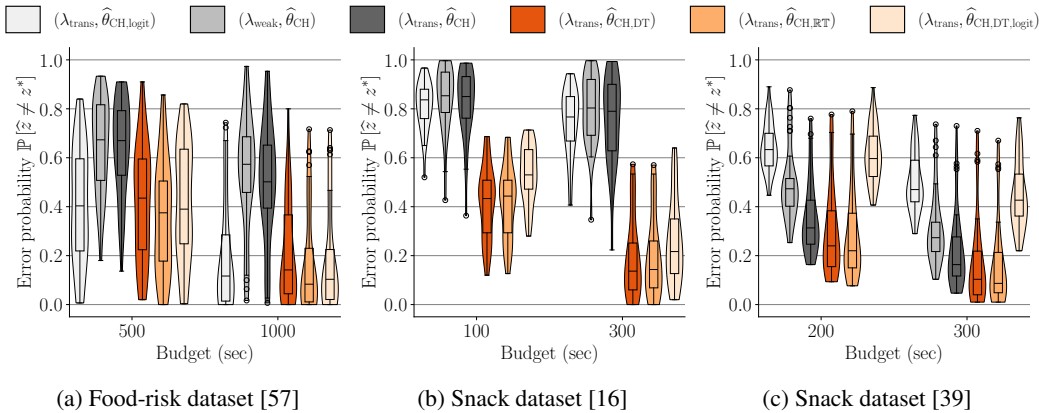

(a) Food-risk dataset [57]    (b) Snack dataset [16]    (c) Snack dataset [39]

Figure 4: This figure shows violin plots (with overlaid box plots) for datasets (a), (b), and (c), showing the distribution of best-arm identification error probabilities, $\mathbb{P}\left[\widehat{z} \neq z^*\right]$, for all bandit instances across six GSE variations and two budgets. The box plots follow the convention of the `matplotlib` Python package. For each GSE variation and budget, the horizontal line in the middle of the box represents the median of the error probabilities across all bandit instances. Each error probability is averaged over 300 repeated simulations under different random seeds. The box's upper and lower borders represent the third and first quartiles, respectively, with whiskers extending to the farthest points within $1.5\times$ the interquartile range. Flier points indicate outliers beyond the whiskers.

The 6th GSE variation, $(\lambda_{\text{trans}}, \widehat{\theta}_{\text{CH,DT,logit}})$, is based on Xiang Chiong et al. [73, eq. (6)], which states that $x^\top \theta^* = \operatorname{sgn}(c_x) \sqrt{\mathbb{E}\left[c_x\right]/\mathbb{E}\left[t_x\right] \cdot 0.5\, \mu^{-1}\left(\mathbb{P}\left[c_x = 1\right]\right)}$. This identity forms the foundation of the estimator in Berlinghieri et al. [8, eq. (7)]. By incorporating our linear utility structure, we obtain the following choice-decision-time estimator $\widehat{\theta}_{\text{CH,DT,logit}}$:

$$
\widehat{\theta}_{\text{CH,DT,logit}} := \left(\sum_{x \in \mathcal{X}_{\text{sample}}} n_x\, xx^\top\right)^{-1} \sum_{x \in \mathcal{X}_{\text{sample}}} n_x\, x \cdot \operatorname{sgn}(c_x) \sqrt{\frac{\mathbb{E}\left[c_x\right]}{\mathbb{E}\left[t_x\right]} \cdot \frac{1}{2}\, \mu^{-1}\left(\widehat{\mathfrak{c}}_x\right)}.
$$

We evaluate six GSE variations on bandit instances constructed from three real-world datasets of human choices and response times. Each dataset includes multiple participants. For each participant, we estimated dEZDM parameters, built a bandit instance, and simulated the GSE variations to assess performance. Details on experimental procedures are provided in appendix D. Key results for the three domains are shown in fig. 4, with full results in appendix D. First, $(\lambda_{\text{trans}}, \widehat{\theta}_{\text{CH,DT}})$ consistently outperforms $(\lambda_{\text{trans}}, \widehat{\theta}_{\text{CH}})$, demonstrating the benefit of incorporating decision times. Second, both of these variations outperform $(\lambda_{\text{weak}}, \widehat{\theta}_{\text{CH}})$, as discussed in section 5.1. Third, $(\lambda_{\text{trans}}, \widehat{\theta}_{\text{CH,DT}})$ performs similarly to $(\lambda_{\text{trans}}, \widehat{\theta}_{\text{CH,RT}})$, suggesting that not knowing the non-decision time has minimal impact. Finally, $(\lambda_{\text{trans}}, \widehat{\theta}_{\text{CH,logit}})$ [67] and $(\lambda_{\text{trans}}, \widehat{\theta}_{\text{CH,DT,logit}})$ [73] do not perform as consistently well as $(\lambda_{\text{trans}}, \widehat{\theta}_{\text{CH,DT}})$, highlighting the effectiveness of our proposed choice-decision-time estimator (eq. (3)).

## 6  Conclusion and future work

This work is the first to leverages human response times to improve fixed-budget best-arm identification in preference-based linear bandits. We proposed a utility estimator that combines choices and response times. Both theoretical and empirical analyses show that response times provide complementary information about preference strength, particularly for queries with strong preferences, enhancing estimation performance. When integrated into a bandit algorithm, incorporating response times consistently improved results across three real-world datasets.

One limitation of this approach is its reliance on reliable response time data, which may be challenging in crowdsourcing settings where participants' focus can vary [45]. Future work could integrate eye-tracking data into the DDM framework [26, 38, 39, 57, 76] to monitor attention and filter unreliable responses. Another direction is to relax the assumption of known non-decision times by estimating them directly from data, following methods proposed by Wagenmakers et al. [67].

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

# A  Broader impacts

Incorporating human response times in human-interactive AI systems provides significant benefits, such as efficiently eliciting user preferences, reducing cognitive loads on users, and improving accessibility for users with disabilities and various cognitive abilities. These benefits can greatly improve recommendation systems, assistive robots, online shopping platforms, and fine-tuning for large language models. However, using human response times also raises concerns about privacy, manipulation, and bias against individuals with slower response times. Governments and law enforcement should work together to mitigate these negative consequences by establishing ethical standards and regulations. Businesses should always obtain user consent before recording response times.

# B   Literature review

## B.1   Bounded accumulation models for choices and response times

Bounded Accumulation Models (BAMs) describe human decision-making using an accumulator (or sampling rule) and a stopping rule [70]. In binary choice tasks, such as two-alternative forced choice tasks, a widely used BAM is the drift-diffusion model (DDM) [51], which models decisions as Brownian motion with fixed boundaries. To capture differences in human response times for correct and incorrect answers, Ratcliff and McKoon [51] allows drift, starting point, and non-decision time to vary across trials. Wagenmakers et al. [67] later introduced the EZ-diffusion model (EZDM), a simplified version of DDM with closed-form solutions for choice and response time moments, making parameter estimation easier and more robust. EZDM assumes deterministic drift, starting point, and non-decision time, fixed across trials, with the starting point equidistant from the boundaries. Berlinghieri et al. [8] specialized EZDM to the difference-based EZDM (dEZDM), where the drift represents the utility difference between two options. For binary queries with arms $z_1$ and $z_2$, the drift is modeled as $u_{z_1} - u_{z_2}$, where $u_{z_1}$ and $u_{z_2}$ are the utilities of $z_1$ and $z_2$.

As discussed in section 2, we impose a linear utility structure on the dEZDM, where each arm's utility is given by $u_z = z^\top \theta^*$, with $\theta^*$ denotes the human preference vector. This approach is supported by both bandit and psychology literature. In bandits, linear utility models scale efficiently with a large number of arms [15, 41]. In psychology, linear combinations of attributes are commonly used in multi-attribute decision-making models [26, 64, 76]. The standard dEZDM in [8, Definition 1] is a special case of our dEZDM with a linear utility structure, where arms correspond to the standard basis vectors in Euclidean space $\mathbb{R}^d$. This mirrors the relationship between multi-armed bandits and linear bandits.

Similarly to our approach, Shvartsman et al. [55] parameterize the human utility function as a Gaussian process and propose a moment-matching Bayesian inference method that uses both choices and response times to estimate latent utilities. Unlike our work, their focus is solely on estimation and does not address bandit optimization. Integrating their estimation techniques into bandit optimization presents an interesting avenue for future research.

Another widely used BAM is the race model [11, 66], which naturally extends to queries with more than two options. In race models, each option has its own accumulator, and the decision ends when any accumulator reaches its barrier. BAMs can also model human attention during decision-making. For example, the attentional-DDM [38, 39, 76] jointly models choices, response times, and eye movements across different options or attributes. Similarly, Thomas et al. [62] introduce the gaze-weighted linear accumulator model to study gaze bias at the trial level. To incorporate learning effects, Pedersen et al. [49] combine reinforcement learning (RL) with DDM, where the human adjusts the drift through RL. In contrast, our work uses RL for AI decision-making when interacting with humans. BAMs also connect to Bayesian RL models of human cognition. For example, Fudenberg et al. [27] propose a model where humans balance decision accuracy and time cost, showing it is equivalent to a DDM with time-decaying boundaries. Neurophysiological evidence supports BAMs. For instance, EEG recordings demonstrate that neurons exhibit accumulation processes and decision thresholds [70]. Additionally, diffusion processes have been used to model neural firing rates [53].

## B.2   Parameter estimation for bounded accumulation models

BAMs often lack closed-form density functions, so hierarchical Bayesian inference is commonly used for parameter estimation [71]. While flexible, these methods are computationally intensive, making them impractical for real-time applications in online learning systems. Faster estimators [8, 67, 73] usually estimate parameters for individual option pairs without leveraging data across pairs. To address this, we propose a computationally efficient method for estimating linear human utility functions, which we integrate into bandit learning. In section 5.2, we empirically show that our estimator outperforms those from prior work [67, 73].

In practice, using response time data requires pre-processing and model fitting, as outlined by Myers et al. [45]. Additionally, Alós-Ferrer et al. [2], Baldassi et al. [6], Fudenberg et al. [28] propose statistical tests to assess the suitability of various DDM extensions for a given dataset.

### B.3  Uses of response times

Response times serve multiple purposes, as highlighted by Clithero [17]. A primary use is improving choice prediction. For instance, Clithero [16] showed that DDM predicts choice probabilities more accurately than the logit model, with parameters estimated through Bayesian Markov chain Monte Carlo. Similarly, Alós-Ferrer et al. [2] demonstrated that response times enhance the identifiability of human preferences compared to using choices alone.

Response times also shed light on human decision-making processes. Castro et al. [14] applied DDM analysis to explore how cognitive workload, induced by secondary tasks, influences decision-making. Analyzing response times has been a long-standing method in cognitive testing to assess mental capabilities [19]. Additionally, Zhang et al. [78, 79] introduced a framework that uses human planning time to infer their intended goals.

Response times can also enhance AI decision-making. In dueling bandits and preference-based RL [7], human choice models are commonly used for preference elicitation. One such model, the random utility model, can be derived from certain BAMs [2]. For example, as discussed after eq. (1), both the Bradley-Terry model [10] and dEZDM [8, 67] yield logistic choice probabilities in the form $\mathbb{P}[z_1 \succ z_2] = \sigma_{logistic}(u_{z_1} - u_{z_2}) = 1/(1 + \exp(-c \cdot (u_{z_1} - u_{z_2})))$, where $u_{z_1}$ and $u_{z_2}$ denote the utilities of $z_1$ and $z_2$ and $c$ is some constant [7, section 3.2]. Our work leverages this connection between random utility models and choice-response-time models to estimate human utilities using both choices and response times.

We hypothesize that our key insight, that response times provide complementary information, especially for queries with strong preferences, extends beyond the dEZDM and the specific logistic link function $\sigma_{logistic}$. Many psychological models capture both choices and response times but lack closed-form choice distributions. In such cases, the choice probability is often expressed as $\mathbb{P}[z_1 \succ z_2] = \sigma^{\dagger}(u_{z_1}, u_{z_2})$, where $\sigma^{\dagger}$ is a function of $u_{z_1}$ and $u_{z_2}$ without a closed form. Fixing $u_{z_2}$ and varying $u_{z_1}$ defines the psychometric function $\sigma^{\dagger}(\cdot, u_{z_2})$, which typically exhibits an "S" shape [60, fig. 1.1]. As preferences become stronger, $\sigma^{\dagger}$ flattens, similar to figs. 1b and 1c, suggesting that choices carry less information. We conjecture that response times remain a valuable complementary signal in such cases.

If we further assume the choice probability depends only on the utility difference, $u_{z_1} - u_{z_2}$, then $\mathbb{P}[z_1 \succ z_2] = \sigma^{\ddagger}(u_{z_1} - u_{z_2})$, where the link function $\sigma^{\ddagger}$ is typically assumed to be strictly monotonic and bounded within $[0, 1]$ [7, section 3.2]. These properties naturally produce an "S"-shaped curve that flattens as preferences become stronger, again suggesting that choices provide less information. In such cases, we conjecture that response times can complement choices to enhance learning.

In summary, BAMs, like DDMs and race models, offer a strong theoretical framework for understanding human decision-making, supported by both behavioral and neurophysiological evidence. These models have been widely applied to choice prediction and the study of human cognitive processes. Our work connects BAMs with bandit algorithms by introducing a computationally efficient estimator for online preference learning. Future research could explore other BAM variants to further examine the benefits of incorporating response times.

## C Proofs

### C.1 Parameters of the difference-based EZ-Diffusion Model (dEZDM) [8, 67]

Given a human preference vector $\theta^*$, for each query $x \in \mathcal{X}$, the utility difference is defined as $u_x := x^\top \theta^*$. In the dEZDM model (introduced in section 2), with barrier $a$, according to Wagenmakers et al. [67, eq. (4), (6), and (9)], the human choice $c_x$ has the following properties:

$$\mathbb{P}\left(c_x = 1\right) = \frac{1}{1 + \exp\left(-2au_x\right)}, \quad \mathbb{P}\left(c_x = -1\right) = \frac{\exp\left(-2au_x\right)}{1 + \exp\left(-2au_x\right)}.$$

Thus, the expected choice is $\mathbb{E}\left[c_x\right] = \tanh(au_x)$, and the choice variance is $\mathbb{V}\left[c_x\right] = 1 - \tanh(au_x)^2$ (restating eq. (1)).

The human decision time $t_x$ has the following properties:

$$\mathbb{E}\left[t_x\right] = \begin{cases} \frac{a}{u_x} \tanh(au_x) & \text{if } u_x \neq 0 \\ a^2 & \text{if } u_x = 0 \end{cases} \quad \text{(restating eq. (1))},$$

$$\mathbb{V}\left[t_x\right] = \begin{cases} \frac{a}{u_x{}^3} \frac{\exp(4au_x) - 1 - 4au_x \exp(2au_x)}{(\exp(2au_x) + 1)^2} & \text{if } u_x \neq 0 \\ 2a^4/3 & \text{if } u_x = 0 \end{cases}.$$

From this, we obtain the following key relationship:

$$\frac{\mathbb{E}\left[c_x\right]}{\mathbb{E}\left[t_x\right]} = \frac{u_x}{a} = x^\top \left(\frac{1}{a}\theta^*\right) \quad \text{(restating eq. (2))}.$$

All these parameters depend solely on the utility difference $u_x := x^\top \theta^*$ and the barrier $a$.

### C.2 Asymptotic normality of the choice-decision-time estimator for estimating the human preference vector $\theta^*$

We now present the proof of the asymptotic normality result for the choice-decision-time estimator, $\widehat{\theta}_{\text{CH,DT}}$, as stated in theorem 3.1, which is restated as follows:

**Theorem 3.1** (Asymptotic normality of $\widehat{\theta}_{\text{CH,DT}}$). *Given a fixed i.i.d. dataset $\left\{x, c_{x,s_{x,i}}, t_{x,s_{x,i}}\right\}_{i \in [n]}$ for each $x \in \mathcal{X}_{sample}$, where $\sum_{x \in \mathcal{X}_{sample}} xx^\top \succ 0$, and assuming that the datasets for different $x \in \mathcal{X}_{sample}$ are independent, then, for any vector $y \in \mathbb{R}^d$, as $n \to \infty$, the following holds:*

$$\sqrt{n}\, y^\top \left(\widehat{\theta}_{CH,DT,n} - \theta^*/a\right) \xrightarrow{D} \mathcal{N}(0, \zeta^2/a^2).$$

*Here, the asymptotic variance depends on a problem-specific constant, $\zeta^2$, with an upper bounded:*

$$\zeta^2 \leq \|y\|^2_{\left(\sum_{x \in \mathcal{X}_{sample}} \left[\min_{x' \in \mathcal{X}_{sample}} \mathbb{E}[t_{x'}]\right] \cdot xx^\top\right)^{-1}}.$$

*Proof.* To simplify notation, we define:

$$\widehat{\mathcal{C}}_x = \frac{1}{n}\sum_{i=1}^{n} c_{x,s_{x,i}}, \quad \mathcal{C}_x = \mathbb{E}\left[c_x\right], \quad \widehat{\mathcal{T}}_x = \frac{1}{n}\sum_{i=1}^{n} t_{x,s_{x,i}}, \quad \mathcal{T}_x = \mathbb{E}\left[t_x\right]. \tag{7}$$

For brevity, we abbreviate $\mathcal{X}_{\text{sample}}$ as $\mathcal{X}$ and $\widehat{\theta}_{\text{CH,DT},n}$ as $\widehat{\theta}$. The estimator $\widehat{\theta}$ can be expressed as:

$$\widehat{\theta} = \left(\sum_{x' \in \mathcal{X}} nx'x'^\top\right)^{-1} \sum_{x \in \mathcal{X}} nx \frac{\widehat{\mathcal{C}}_x}{\widehat{\mathcal{T}}_x} \quad \text{(restating eq. (3))}.$$

We rewrite $\theta^*/a$ as:

$$\begin{aligned} \theta^*/a &= \left(\sum_{x' \in \mathcal{X}} nx'x'^\top\right)^{-1} \sum_{x \in \mathcal{X}} nxx^\top \frac{\theta^*}{a} \\ &= \left(\sum_{x' \in \mathcal{X}} nx'x'^\top\right)^{-1} \sum_{x \in \mathcal{X}} nx \frac{\mathcal{C}_x}{\mathcal{T}_x}. \end{aligned} \tag{8}$$

Therefore, for any vector $y \in \mathbb{R}^d$, we have:

$$y^\top \left(\widehat{\theta} - \frac{\theta^*}{a}\right) = y^\top \left(\sum_{x' \in \mathcal{X}} n x' x'^\top\right)^{-1} \sum_{x \in \mathcal{X}} nx \left(\frac{\widehat{\mathcal{C}}_x}{\widehat{\mathcal{T}}_x} - \frac{\mathcal{C}_x}{\mathcal{T}_x}\right) =: \sum_{x \in \mathcal{X}} \xi_x \left(\frac{\widehat{\mathcal{C}}_x}{\widehat{\mathcal{T}}_x} - \frac{\mathcal{C}_x}{\mathcal{T}_x}\right), \quad (9)$$

where $\xi_x$ is defined as $\xi_x := y^\top \left(\sum_{x' \in \mathcal{X}} n x' x'^\top\right)^{-1} nx$. In eq. (9), the only random variables are $\widehat{\mathcal{C}}_x$ and $\widehat{\mathcal{T}}_x$. For simplicity, for any $x_i \in \mathcal{X} := \{x_1, \cdots, x_{|\mathcal{X}|}\}$, we slighly abuse the notation and use $\xi_i, c_i, t_i, \mathcal{C}_i, \mathcal{T}_i, \widehat{\mathcal{C}}_i$ and $\widehat{\mathcal{T}}_i$ denote $\xi_{x_i}, c_{x_i}, t_{x_i}, \mathcal{C}_{x_i}, \mathcal{T}_{x_i}, \widehat{\mathcal{C}}_{x_i}$, and $\widehat{\mathcal{T}}_{x_i}$, respectively. By applying the multidimensional central limit theorem, we have:

$$\sqrt{n} \begin{bmatrix} \widehat{\mathcal{C}}_1 - \mathcal{C}_1 \\ \widehat{\mathcal{T}}_1 - \mathcal{C}_1 \\ \vdots \\ \widehat{\mathcal{C}}_{|\mathcal{X}|} - \mathcal{C}_{|\mathcal{X}|} \\ \widehat{\mathcal{T}}_{|\mathcal{X}|} - \mathcal{C}_{|\mathcal{X}|} \end{bmatrix} \xrightarrow{D} \mathcal{N}\left(0, \begin{bmatrix} \begin{matrix} \mathbb{V}[c_1] & \text{cov}[c_1, t_1] \\ \text{cov}[t_1, c_1] & \mathbb{V}[t_1] \end{matrix} & & \\ & \ddots & \\ & & \begin{matrix} \mathbb{V}[c_{|\mathcal{X}|}] & \text{cov}[c_{|\mathcal{X}|}, t_{|\mathcal{X}|}] \\ \text{cov}[t_{|\mathcal{X}|}, c_{|\mathcal{X}|}] & \mathbb{V}[t_{|\mathcal{X}|}] \end{matrix} \end{bmatrix}\right)$$

$$= \mathcal{N}\left(0, \text{diag}\left[\mathbb{V}[c_1], \mathbb{V}[t_1], \cdots, \mathbb{V}[c_{|\mathcal{X}|}], \mathbb{V}[t_{|\mathcal{X}|}]\right]\right). \quad (10)$$

In the first line of eq. (10), the block-diagonal structure of the covariance matrix emerges because $(\widehat{\mathcal{C}}_i, \widehat{\mathcal{T}}_i)_{i \in [|\mathcal{X}|]}$ are independent of each other. For any fixed $x_i$, to derive the second line of eq. (10), we use the fact that:

$$\mathbb{E}[t_i c_i] = \mathbb{P}(c_i = 1)\mathbb{E}[1 \cdot t_i | c_i = 1] + \mathbb{P}(c_i = -1)\mathbb{E}[-1 \cdot t_i | c_i = -1]$$
$$\overset{(i)}{=} (\mathbb{P}(c_i = 1) - \mathbb{P}(c_i = -1))\mathbb{E}[t_i | c_i = 1] \quad (11)$$
$$= \mathbb{E}[c_i]\mathbb{E}[t_i],$$

where $(i)$ is because $\mathbb{E}[t_i|c_i = 1] = \mathbb{E}[t_i|c_i = -1]$ [48, eq. (A.7) and (A.9)]. Therefore, eq. (11) implies that $\text{cov}(c_i, t_i) = 0$ [3], which justifies the second line of eq. (10).

Now, let us define the function $g(c_1, t_1, \cdots, c_{|\mathcal{X}|}, t_{|\mathcal{X}|}) := \sum_{i \in [|\mathcal{X}|]} \xi_i c_i / t_i$. The gradient of $g$ is:

$$\nabla g|_{(c_1, t_1, \cdots, c_{|\mathcal{X}|}, t_{|\mathcal{X}|})} = \begin{bmatrix} \xi_1/t_1 & -\xi_1 c_1/t_1^2 & \cdots & \xi_{|\mathcal{X}|}/t_{|\mathcal{X}|} & -\xi_{|\mathcal{X}|} c_{|\mathcal{X}|}/t_{|\mathcal{X}|}^2 \end{bmatrix}^\top. \quad (12)$$

Using the multivariate delta method, we obtain:

$$\sqrt{n} \sum_{i \in [|\mathcal{X}|]} \xi_i \left(\frac{\widehat{\mathcal{C}}_i}{\widehat{\mathcal{T}}_i} - \frac{\mathcal{C}_i}{\mathcal{T}_i}\right)$$

$$= \sqrt{n}\left(g\left(\widehat{\mathcal{C}}_1, \widehat{\mathcal{T}}_1, \cdots, \widehat{\mathcal{C}}_{|\mathcal{X}|}, \widehat{\mathcal{T}}_{|\mathcal{X}|}\right) - g\left(\mathcal{C}_1, \mathcal{T}_1, \cdots, \mathcal{C}_{|\mathcal{X}|}, \mathcal{T}_{|\mathcal{X}|}\right)\right)$$

$$\xrightarrow{D} \mathcal{N}\left(0, \nabla g^\top|_{(\mathcal{C}_1, \mathcal{T}_1, \cdots, \mathcal{C}_{|\mathcal{X}|}, \mathcal{T}_{|\mathcal{X}|})} \begin{bmatrix} \mathbb{V}[c_1] & & & \\ & \mathbb{V}[t_1] & & \\ & & \ddots & \\ & & & \mathbb{V}[c_{|\mathcal{X}|}] \\ & & & & \mathbb{V}[t_{|\mathcal{X}|}] \end{bmatrix} \nabla g|_{(\mathcal{C}_1, \mathcal{T}_1, \cdots, \mathcal{C}_{|\mathcal{X}|}, \mathcal{T}_{|\mathcal{X}|})}\right)$$

$$= \mathcal{N}\left(0, \sum_{i \in [|\mathcal{X}|]} \xi_i^2 \left(\frac{1}{\mathcal{T}_i^2}\mathbb{V}(c_i) + \frac{\mathcal{C}_i^2}{\mathcal{T}_i^4}\mathbb{V}(t_i)\right)\right)$$

$$= \mathcal{N}\left(0, \frac{1}{a^2} \sum_{i \in [|\mathcal{X}|]} \xi_i^2 \left(\frac{a^2}{\mathcal{T}_i^2}\mathbb{V}(c_i) + \frac{a^2 \mathcal{C}_i^2}{\mathcal{T}_i^4}\mathbb{V}(t_i)\right)\right) \quad (13)$$

---

[3] Equation (11) implies that for any query $x_i$, the human choice $c_i$ and decision time $t_i$ are uncorrelated. Moreover, they are independent, as discussed by Drugowitsch [22, the discussion above eq. (7)] and Baldassi et al. [6, proposition 3].

By applying the identities outlined in appendix C.1, we can establish the following identity:

$$\forall i \in [|\mathcal{X}|] \colon \frac{a^2}{\mathcal{T}_i^2} \mathbb{V}(c_i) + \frac{a^2 \mathcal{C}_i^2}{\mathcal{T}_i^4} \mathbb{V}(t_i) = \frac{1}{\mathcal{T}_i}. \tag{14}$$

Substituting this identity into eq. (13), we obtain:

$$\sqrt{n} \sum_{i \in [|\mathcal{X}|]} \xi_i \left( \frac{\widehat{\mathcal{C}_i}}{\widehat{\mathcal{T}_i}} - \frac{\mathcal{C}_i}{\mathcal{T}_i} \right) \xrightarrow{D} \mathcal{N} \left( 0, \frac{1}{a^2} \sum_{i \in [|\mathcal{X}|]} \xi_i^2 \frac{1}{\mathcal{T}_i} \right). \tag{15}$$

Finally, the asymptotic variance can be upper bounded as follows:

$$
\begin{aligned}
&\frac{1}{a^2} \sum_{i \in [|\mathcal{X}|]} \xi_i^2 \frac{1}{\mathcal{T}_i} \\
\leq{}& \frac{1}{a^2} \frac{1}{\min_{i \in [|\mathcal{X}|]} \mathcal{T}_i} \sum_{i \in [|\mathcal{X}|]} \xi_i^2 \\
={}& \frac{1}{a^2} \frac{1}{\min_{i \in [|\mathcal{X}|]} \mathcal{T}_i} \cdot \left( \sum_{x \in \mathcal{X}} y^\top \left( \sum_{x' \in \mathcal{X}} n x' x'^\top \right)^{-1} n^2 x x^\top \left( \sum_{x' \in \mathcal{X}} n x' x'^\top \right)^{-1} y \right) \\
={}& \frac{1}{a^2} \frac{1}{\min_{i \in [|\mathcal{X}|]} \mathcal{T}_i} \cdot y^\top \left( \sum_{x' \in \mathcal{X}} x' x'^\top \right)^{-1} y \\
={}& \frac{1}{a^2} y^\top \left( \sum_{x' \in \mathcal{X}} \left[ \min_{i \in [|\mathcal{X}|]} \mathcal{T}_i \right] x' x'^\top \right)^{-1} y \\
\equiv{}& \frac{1}{a^2} \|y\|^2_{\left( \sum_{x' \in \mathcal{X}} \left[ \min_{i \in [|\mathcal{X}|]} \mathcal{T}_i \right] x' x'^\top \right)^{-1}}.
\end{aligned}
\tag{16}
$$

$\square$

### C.3 Non-asymptotic concentration of the two estimators for estimating the utility difference $u_x$ given a query $x$

#### C.3.1 The choice-decision-time estimator

Section 3.3 focuses on the problem of estimating the utility difference for a single query. Given a query $x \in \mathcal{X}$, the objective is to estimate the utility difference $u_x := x^\top \theta^*$ using an i.i.d. dataset, denoted by $\left\{ \left( c_{x,s_{x,i}}, t_{x,s_{x,i}} \right) \right\}_{i \in [n_x]}$.

We begin by applying the choice-decision-time estimator from eq. (3), which is derived by solving the following least squares problem:

$$\widehat{\theta}_{\text{CH,DT}} = \arg\min_{\theta \in \mathbb{R}^d} \sum_{x \in \mathcal{X}_{\text{sample}}} n_x \left( x^\top \theta - \frac{\sum_{i \in [n_x]} c_{x,s_{x,i}}}{\sum_{i \in [n_x]} t_{x,s_{x,i}}} \right)^2 .$$

Similarly, the utility difference for a single query is estimated as the solution to the following least squares problem, yielding the estimate:

$$\widehat{u}_{x,\text{CH,DT}} = \arg\min_{u \in \mathbb{R}} \left( u - \frac{\sum_{i \in [n_x]} c_{x,s_{x,i}}}{\sum_{i \in [n_x]} t_{x,s_{x,i}}} \right)^2 = \frac{\sum_{i \in [n_x]} c_{x,s_{x,i}}}{\sum_{i \in [n_x]} t_{x,s_{x,i}}} \quad \text{(restating eq. (5))}.$$

The resulting estimate, $\widehat{u}_{x,\text{CH,DT}}$, approximates $u_x/a$ rather than $u_x$. However, since the ranking of arm utilities is preserved between $u_x/a$ and $u_x$, estimating $u_x/a$ is sufficient for the purpose of best-arm identification.

For the case where the utility difference $u_x \neq 0$, the non-asymptotic concentration inequality for this estimator is presented in theorem 3.3. To prove this, we first introduce lemma C.1, which demonstrates that for any given query $x$, the decision time is a sub-exponential random variable.

To simplify notation, we define:

$$\widehat{\mathcal{C}}_x = \frac{1}{n_x} \sum_{i=1}^{n_x} c_{x,s_{x,i}}, \quad \mathcal{C}_x = \mathbb{E}\left[ c_x \right], \quad \widehat{\mathcal{T}}_x = \frac{1}{n_x} \sum_{i=1}^{n_x} t_{x,s_{x,i}}, \quad \mathcal{T}_x = \mathbb{E}\left[ t_x \right], \quad \widehat{u}_{x,\text{CH,DT}} = \frac{\widehat{\mathcal{C}}_x}{\widehat{\mathcal{T}}_x}. \tag{17}$$

**Lemma C.1.** If $u_x \neq 0$, then $(t_x - \mathcal{T}_x)$ is sub-exponential SE $\left( \nu_x^2, \alpha_x \right)$, where $\nu_x = \sqrt{2}a/|u_x|$ and $\alpha_x = 2/u_x^2$.

*Proof.* For simplicity, we will omit the subscript $x$ throughout the proof and assume, without loss of generality, that $u > 0$.

Our objective is to establish the following inequality, which holds for all $s \in (-u^2/2, u^2/2)$:

$$\mathbb{E}\left( \exp\left( s\left( t - \mathcal{T} \right) \right) \right) \leq \exp\left( \frac{2a^2/u^2}{2} s^2 \right). \tag{18}$$

This implies that $(t - \mathcal{T})$ is sub-exponential SE $\left( \nu^2, \alpha \right)$, as defined by Wainwright [69, Definition 2.7].

**Step 1: Transform eq. (18) into a more manageable inequality (eq. (24)).**

Using Cox [18, eq. (128)], with $\Delta := u^2 - 2s$, $\theta_1 := -u - \sqrt{\Delta}$ and $\theta_2 := -u + \sqrt{\Delta}$, we have[4]:

$$
\begin{aligned}
\mathbb{E}\left(\exp\left(st\right)\right) &= \frac{\exp\left(a\theta_1\right) - \exp\left(2a\theta_2 + a\theta_1\right)}{\exp\left(2a\theta_1\right) - \exp\left(2a\theta_2\right)} - \frac{\exp\left(a\theta_2\right) - \exp\left(2a\theta_1 + a\theta_2\right)}{\exp\left(2a\theta_1\right) - \exp\left(2a\theta_2\right)} \\
&= \frac{\exp\left(a\theta_1\right)\left[1 + \exp\left(a\theta_1 + a\theta_2\right)\right]}{\exp\left(2a\theta_1\right) - \exp\left(2a\theta_2\right)} - \frac{\exp\left(a\theta_2\right)\left[1 + \exp\left(a\theta_2 + a\theta_1\right)\right]}{\exp\left(2a\theta_1\right) - \exp\left(2a\theta_2\right)} \\
&= \frac{\left[\exp\left(a\theta_1\right) - \exp\left(a\theta_2\right)\right]\left[1 + \exp\left(a\theta_2 + a\theta_1\right)\right]}{\exp\left(2a\theta_1\right) - \exp\left(2a\theta_2\right)} \\
&= \frac{1 + \exp\left(a\theta_2 + a\theta_1\right)}{\exp\left(a\theta_1\right) + \exp\left(a\theta_2\right)} \\
&= \frac{\exp\left(-au\right) + \exp\left(au\right)}{\exp\left(-a\sqrt{\Delta}\right) + \exp\left(a\sqrt{\Delta}\right)} \\
&=: \frac{N}{D(s)}.
\end{aligned}
\tag{19}
$$

In the last line, we define $N = 2\cosh(au)$ and $D(s) = 2\cosh(a\sqrt{\Delta})$. Thus, we arrive at:

$$
\mathbb{E}\left(\exp\left(s \cdot (t - \mathcal{T})\right)\right) = \frac{N}{D(s)} \cdot \frac{1}{\exp\left(s \cdot \mathcal{T}\right)} = \frac{N}{\exp\left(sa\tanh(au)/u\right) D(s)}.
\tag{20}
$$

To prove the original inequality in eq. (18), it is now sufficient to show:

$$
D(s) \cdot \exp\left(\frac{a}{u}\tanh(au)s + \frac{a^2}{u^2}s^2\right) \geq N.
\tag{21}
$$

For $s = 0$, the inequality holds trivially, as:

$$
D(0) \cdot 1 = 2\cosh(au) = N.
\tag{22}
$$

For $s \neq 0$, taking the derivative of the left-hand side of eq. (21) yields:

$$
\begin{aligned}
&\frac{\mathrm{d}}{\mathrm{d}s}\left(D(s) \cdot \exp\left(\frac{a}{u}\tanh(au)s + \frac{a^2}{u^2}s^2\right)\right) \\
&= \exp\left(\frac{a}{u}\tanh(au)s + \frac{a^2}{u^2}s^2\right) \cdot \left(-\frac{2a}{\sqrt{\Delta}}\sinh\left(a\sqrt{\Delta}\right) + 2\cosh\left(a\sqrt{\Delta}\right) \cdot \left(\frac{a}{u}\tanh(au) + 2\frac{a^2}{u^2}s\right)\right) \\
&= 2\exp\left(\frac{a}{u}\tanh(au)s + \frac{a^2}{u^2}s^2\right)\cosh\left(a\sqrt{\Delta}\right) \cdot \left(-\frac{a}{\sqrt{\Delta}}\tanh\left(a\sqrt{\Delta}\right) + \frac{a}{u}\tanh(au) + 2\frac{a^2}{u^2}s\right).
\end{aligned}
\tag{23}
$$

In step 2, we will prove the following inequality:

$$
-\frac{a}{\sqrt{\Delta}}\tanh\left(a\sqrt{\Delta}\right) + \frac{a}{u}\tanh(au) + 2\frac{a^2}{u^2}s \begin{cases} \geq 0, & \forall s \geq 0, \\ < 0, & \forall s < 0, \end{cases}
\tag{24}
$$

Equation (24) implies that $D(s) \cdot \exp\left(\frac{a}{u}\tanh(au)s + \frac{a^2}{u^2}s^2\right) \geq N$, which finishes the proof.

**Step 2. Prove eq. (24).**

---

[4]In Cox [18, eq. (128)], setting $a = 2a$ and $x_0 = a$ leads to the desired result.

For $s \geq 0$, the following holds:

$$-\frac{a}{\sqrt{\Delta}} \tanh\left(a\sqrt{\Delta}\right) + \frac{a}{u} \tanh(au) + 2\frac{a^2}{u^2}s$$

$$\overset{(i)}{\geq} a \tanh\left(a\sqrt{\Delta}\right)\left(\frac{1}{u} - \frac{1}{\sqrt{\Delta}}\right) + 2\frac{a^2}{u^2}s$$

$$= a \tanh\left(a\sqrt{\Delta}\right)\frac{-2s}{u\sqrt{\Delta}\left(\sqrt{\Delta}+u\right)} + 2\frac{a^2}{u^2}s$$

$$= -2s \cdot \frac{a^2}{u\left(\sqrt{\Delta}+u\right)} \cdot \frac{\tanh\left(a\sqrt{\Delta}\right)}{a\sqrt{\Delta}} + 2\frac{a^2}{u^2}s \qquad (25)$$

$$\overset{(ii)}{\geq} -2s\frac{a^2}{u^2}\cdot 1 + 2\frac{a^2}{u^2}s$$

$$= 0.$$

Here, $(i)$ follows from $\tanh(au) \geq \tanh(a\sqrt{\Delta}) = \tanh(a\sqrt{u^2-2s})$ and $(ii)$ follows from $\tanh(x)/x \leq 1$.

For $s < 0$, the following holds:

$$-\frac{a}{\sqrt{\Delta}} \tanh\left(a\sqrt{\Delta}\right) + \frac{a}{u} \tanh(au) + 2\frac{a^2}{u^2}s$$

$$\overset{(i)}{\leq} a \tanh\left(a\sqrt{\Delta}\right)\left(\frac{1}{u} - \frac{1}{\sqrt{\Delta}}\right) + 2\frac{a^2}{u^2}s$$

$$= -2s \cdot \frac{a^2}{u\left(\sqrt{\Delta}+u\right)} \cdot \frac{\tanh\left(a\sqrt{\Delta}\right)}{a\sqrt{\Delta}} + 2\frac{a^2}{u^2}s \qquad (26)$$

$$\overset{(ii)}{\leq} -2s\frac{a^2}{u^2}\cdot 1 + 2\frac{a^2}{u^2}s$$

$$= 0.$$

Here, $(i)$ follows from $\tanh(au) \leq \tanh(a\sqrt{\Delta}) = \tanh(a\sqrt{u^2-2s})$ and $(ii)$ follows from $\tanh(x)/x \leq 1$.

By combining both cases, we conclude that the inequality in eq. (24) holds, which completes Step 2 and proves the desired result. $\square$

Next, we prove theorem 3.3, which provides the non-asymptotic concentration inequality for the estimator from eq. (5), restated as follows:

**Theorem 3.3** (Non-asymptotic concentration of $\widehat{u}_{x,\text{CH,DT}}$). *For each query $x \in \mathcal{X}$ with $u_x \neq 0$, given a fixed i.i.d. dataset $\{(c_{x,s_{x,i}}, t_{x,s_{x,i}})\}_{i\in[n_x]}$, for any $\epsilon > 0$ satisfying $\epsilon \leq \min\{|u_x|/(\sqrt{2}a), (1+\sqrt{2})a|u_x|/\mathbb{E}[t_x]\}$, the following holds:*

$$\mathbb{P}\left(\left|\widehat{u}_{x,\text{CH,DT}} - \frac{u_x}{a}\right| > \epsilon\right) \leq 4\exp\left(-\left[m_{\text{CH,DT}}^{\text{non-asym}}\left(x^\top\theta^*\right)\right]^2 n_x \left[\epsilon \cdot a\right]^2\right),$$

*where $m_{\text{CH,DT}}^{\text{non-asym}}\left(x^\top\theta^*\right) := \mathbb{E}[t_x] / \left[(2+2\sqrt{2})a\right]$.*

*Proof.* For clarity, we will omit the subscripts $x$ throughout this proof. Based on lemma C.1, we define the constants $\nu := \sqrt{2}a/|u|$ and $\alpha := 2/u^2$.

We begin by introducing $\epsilon_\mathcal{C} := \mathcal{T}/\left(\sqrt{2} + \sqrt{2}\nu|\mathcal{C}|/\mathcal{T}\right)\cdot\epsilon$ and $\epsilon_\mathcal{T} := \nu\epsilon_\mathcal{C}$. From the identities provided in appendix C.1, we know that $\nu|\mathcal{C}|/\mathcal{T} = \sqrt{2}a/|u| \cdot |u|/a = \sqrt{2}$. This allows us to simplify the constants $\epsilon_\mathcal{C}$ and $\epsilon_\mathcal{T}$ as:

$$\epsilon_\mathcal{C} = \frac{\mathcal{T}}{\sqrt{2}\left(\sqrt{2}+1\right)}\epsilon \quad \text{and} \quad \epsilon_\mathcal{T} = \frac{\nu\mathcal{T}}{\sqrt{2}\left(\sqrt{2}+1\right)}\epsilon. \qquad (27)$$

For any $\epsilon$ satisfying the following condition:

$$\epsilon \leq \min\left\{\frac{1}{\nu}, \frac{\sqrt{2}(1+\sqrt{2})\nu}{\alpha\mathcal{T}}\right\}, \tag{28}$$

we observe that $\epsilon_{\mathcal{T}} < \min\left\{\mathcal{T}(1 - 1/\sqrt{2}), \nu^2/\alpha\right\}$. We can now apply lemma C.2 to derive the following:

$$\mathbb{P}\left(\left|\widehat{\mathcal{T}} - \mathcal{T}\right| > \epsilon_{\mathcal{T}}\right) \leq 2\exp\left(-\frac{n\epsilon_{\mathcal{T}}^2}{2\nu^2}\right). \tag{29}$$

Thus, by combining the results, we conclude:

$$
\begin{aligned}
\mathbb{P}\left(\left|\frac{\widehat{\mathcal{C}}}{\widehat{\mathcal{T}}} - \frac{\mathcal{C}}{\mathcal{T}}\right| > \epsilon\right) &= \mathbb{P}\left(\left|\frac{\widehat{\mathcal{C}}}{\widehat{\mathcal{T}}} - \frac{\mathcal{C}}{\mathcal{T}}\right| > \sqrt{2}\frac{\epsilon_{\mathcal{C}} + \epsilon_{\mathcal{T}} \cdot |\mathcal{C}|/\mathcal{T}}{\mathcal{T}}\right) \\
&\overset{(i)}{\leq} \mathbb{P}\left(\left|\widehat{\mathcal{C}} - \mathcal{C}\right| > \epsilon_{\mathcal{C}}\right) + \mathbb{P}\left(\left|\widehat{\mathcal{T}} - \mathcal{T}\right| > \epsilon_{\mathcal{T}}\right) \\
&\overset{(ii)}{\leq} 2\exp\left(-\frac{n\epsilon_{\mathcal{C}}^2}{2}\right) + 2\exp\left(-\frac{n\epsilon_{\mathcal{T}}^2}{2\nu^2}\right) \\
&\overset{(iii)}{=} 4\exp\left(-\frac{n\epsilon_{\mathcal{C}}^2}{2}\right) \\
&= 4\exp\left(-\frac{\mathcal{T}^2}{4\left(1+\sqrt{2}\right)^2} \cdot n\epsilon^2\right).
\end{aligned}
\tag{30}
$$

Here, $(i)$ follows from lemma C.3, $(ii)$ uses lemma C.2 and eq. (29), and $(iii)$ follows from eq. (27). $\qquad\square$

## Supporting Details

**Lemma C.2.** *For each query $x$ with $u_x \neq 0$, and constants $\epsilon_{\mathcal{C}} > 0$ and $\epsilon_{\mathcal{T}} \in (0, \nu_x^2/\alpha_x]$, the following inequalities hold:*

$$\mathbb{P}\left(\left|\widehat{\mathcal{C}}_x - \mathcal{C}_x\right| \geq \epsilon_{\mathcal{C}}\right) \leq 2\exp\left(-\frac{n\epsilon_{\mathcal{C}}^2}{2}\right), \quad \mathbb{P}\left(\left|\widehat{\mathcal{T}}_x - \mathcal{T}_x\right| \geq \epsilon_{\mathcal{T}}\right) \leq 2\exp\left(-\frac{n\epsilon_{\mathcal{T}}^2}{2\nu_x^2}\right). \tag{31}$$

*Here, the constants are $\nu_x := \sqrt{2}a/|u_x|$ and $\alpha_x := 2/u_x^2$.*

*Proof.* Since $c_x \in \{-1, 1\}$, by applying Hoeffding's inequality [69, proposition 2.5], we obtain:

$$\mathbb{P}\left(\left|\widehat{\mathcal{C}}_x - \mathcal{C}_x\right| \geq \epsilon_{\mathcal{C}}\right) \leq 2\exp\left(-\frac{n\epsilon_{\mathcal{C}}^2}{2}\right). \tag{32}$$

From lemma C.1, we know that $t_x$ is sub-exponential $SE(\nu_x^2, \alpha_x)$. By applying Wainwright [69, proposition 2.9 and eq. (2.18)], we obtain:

$$\mathbb{P}\left(\left|\widehat{\mathcal{T}}_x - \mathcal{T}_x\right| \geq \epsilon_{\mathcal{T}}\right) \leq 2\exp\left(-\frac{n\epsilon_{\mathcal{T}}^2}{2\nu_x^2}\right), \quad \forall\epsilon_{\mathcal{T}} \in (0, \nu_x^2/\alpha_x]. \tag{33}$$

$\qquad\square$

**Lemma C.3.** *Consider constants $\mathcal{C} \in \mathbb{R}$, $\mathcal{T} > 0$, $\epsilon_{\mathcal{C}} > 0$, and $\epsilon_{\mathcal{T}} \in \left(0, (1 - 1/\sqrt{2})\mathcal{T}\right)$. For any $\widehat{\mathcal{C}} \in [\mathcal{C} - \epsilon_{\mathcal{C}}, \mathcal{C} + \epsilon_{\mathcal{C}}]$ and $\widehat{\mathcal{T}} \in [\mathcal{T} - \epsilon_{\mathcal{T}}, \mathcal{T} + \epsilon_{\mathcal{T}}]$, the following inequality holds*

$$\left|\frac{\widehat{\mathcal{C}}}{\widehat{\mathcal{T}}} - \frac{\mathcal{C}}{\mathcal{T}}\right| \leq \sqrt{2}\frac{\epsilon_{\mathcal{C}} + \epsilon_{\mathcal{T}} \cdot |\mathcal{C}|/\mathcal{T}}{\mathcal{T}}. \tag{34}$$

*Proof.* The maximum value of $\left|\widehat{\mathcal{C}}/\widehat{\mathcal{T}} - \mathcal{C}/\mathcal{T}\right|$ is attained at the extremum of $\widehat{\mathcal{C}}/\widehat{\mathcal{T}}$. Since $\widehat{\mathcal{C}}/\widehat{\mathcal{T}}$ is linear in $\widehat{\mathcal{C}}$, the extremum of $\widehat{\mathcal{C}}/\widehat{\mathcal{T}}$ is attained at $C^* \in \{\mathcal{C} - \epsilon_{\mathcal{C}}, \mathcal{C} + \epsilon_{\mathcal{C}}\}$ for any $\widehat{\mathcal{T}} \in [\mathcal{T} - \epsilon_{\mathcal{T}}, \mathcal{T} + \epsilon_{\mathcal{T}}] > 0$. Given that $\widehat{\mathcal{T}} > 0$, the extremum of $C^*/\widehat{\mathcal{T}}$ is attained at $T^* \in \{\mathcal{T} - \epsilon_{\mathcal{T}}, \mathcal{T} + \epsilon_{\mathcal{T}}\}$. Therefore, the extremum of $\widehat{\mathcal{C}}/\widehat{\mathcal{T}}$ lies in the set:

$$\max_{\substack{\widehat{\mathcal{C}} \in [\mathcal{C} - \epsilon_{\mathcal{C}}, \mathcal{C} + \epsilon_{\mathcal{C}}] \\ \widehat{\mathcal{T}} \in [\mathcal{T} - \epsilon_{\mathcal{T}}, \mathcal{T} + \epsilon_{\mathcal{T}}]}} \frac{\widehat{\mathcal{C}}}{\widehat{\mathcal{T}}} \in \left\{ \frac{\mathcal{C} - \epsilon_{\mathcal{C}}}{\mathcal{T} - \epsilon_{\mathcal{T}}}, \quad \frac{\mathcal{C} - \epsilon_{\mathcal{C}}}{\mathcal{T} + \epsilon_{\mathcal{T}}}, \quad \frac{\mathcal{C} + \epsilon_{\mathcal{C}}}{\mathcal{T} - \epsilon_{\mathcal{T}}}, \quad \frac{\mathcal{C} + \epsilon_{\mathcal{C}}}{\mathcal{T} + \epsilon_{\mathcal{T}}} \right\}. \tag{35}$$

For any combination $(s_{\mathcal{C}}, s_{\mathcal{T}}) \in \{\pm 1\} \times \{\pm 1\}$, and using the function $\epsilon_{\mathcal{T}} \leq (1 - 1/\sqrt{2})\mathcal{T}$, we have:

$$\left| \frac{\mathcal{C} + s_{\mathcal{C}}\epsilon_{\mathcal{C}}}{\mathcal{T} + s_{\mathcal{T}}\epsilon_{\mathcal{T}}} - \frac{\mathcal{C}}{\mathcal{T}} \right| = \left| \frac{s_{\mathcal{C}}\epsilon_{\mathcal{C}}\mathcal{T} - s_{\mathcal{T}}\epsilon_{\mathcal{T}}\mathcal{C}}{\mathcal{T}(\mathcal{T} + s_{\mathcal{T}}\epsilon_{\mathcal{T}})} \right| \leq \frac{\epsilon_{\mathcal{C}}\mathcal{T} + \epsilon_{\mathcal{T}}|\mathcal{C}|}{\mathcal{T}(\mathcal{T} - \epsilon_{\mathcal{T}})} \leq \sqrt{2}\frac{\epsilon_{\mathcal{C}}\mathcal{T} + \epsilon_{\mathcal{T}}|\mathcal{C}|}{\mathcal{T}^2}. \tag{36}$$

By combining these results, we conclude that:

$$\max_{\substack{\widehat{\mathcal{C}} \in [\mathcal{C} - \epsilon_{\mathcal{C}}, \mathcal{C} + \epsilon_{\mathcal{C}}] \\ \widehat{\mathcal{T}} \in [\mathcal{T} - \epsilon_{\mathcal{T}}, \mathcal{T} + \epsilon_{\mathcal{T}}]}} \left| \frac{\widehat{\mathcal{C}}}{\widehat{\mathcal{T}}} - \frac{\mathcal{C}}{\mathcal{T}} \right| = \max_{(s_{\mathcal{C}}, s_{\mathcal{T}}) \in \{\pm 1\} \times \{\pm 1\}} \left| \frac{\mathcal{C} + s_{\mathcal{C}}\epsilon_{\mathcal{C}}}{\mathcal{T} + s_{\mathcal{T}}\epsilon_{\mathcal{T}}} - \frac{\mathcal{C}}{\mathcal{T}} \right| \leq \sqrt{2}\frac{\epsilon_{\mathcal{C}} + \epsilon_{\mathcal{T}}|\mathcal{C}|/\mathcal{T}}{\mathcal{T}}.$$

$\square$

### C.3.2 The choice-only estimator

We now apply the logistic-regression-based choice-only estimator from eq. (4) to estimate the utility difference for a single query. Recall that for each query $x \in \mathcal{X}$, the human choice $c_x \in \{-1, 1\}$. We define the binary-encoded choice as $e_x := (c_x + 1)/2 \in \{0, 1\}$. We reformulate the MLE in eq. (4) into a utility difference estimation problem for a single query, leading to the following optimization problem:

$$\widehat{u}_{x,\text{CH}} = \arg\max_{u \in \mathbb{R}} \sum_{i \in [n_x]} \log \mu(c_{x,s_{x,i}} u)$$

$$= \arg\max_{u \in \mathbb{R}} \sum_{i \in [n_x]} \log \left[ (\mu(u))^{e_{x,s_{x,i}}} \cdot (\mu(-u))^{1 - e_{x,s_{x,i}}} \right].$$

The first-order optimality condition provides the optimal solution:

$$\widehat{u}_{x,\text{CH}} = \mu^{-1} \left( \frac{1}{n_x} \sum_{i \in [n_x]} e_{x,s_{x,i}} \right) \quad \text{(restating eq. (6))},$$

where $\mu^{-1}(p) := \log(p/(1-p))$ is the logit function (also known as the log-odds), defined as the inverse of the function $\mu(\cdot)$ introduced in eq. (4).

The resulting estimate, $\widehat{u}_{x,\text{CH}}$, from eq. (6) gives an estimate of $2au_x$, not $u_x$. However, since the ranking of arm utilities based on $2au_x$ is the same as that based on the true $u_x$, estimating $2au_x$ suffices for identifying the best arm.

The non-asymptotic concentration inequality for this estimator is stated in theorem 3.4. This result is directly adapted from Jun et al. [31, theorem 5], by letting $x_1 = \cdots = x_t = 1$ and $t_{\text{eff}} = d = 1$.

# D  Experiment details

Our empirical experiments (Sec. 5) were conducted on a MacBook Pro (M3 Pro, Nov 2023) with 36 GB of memory.

Our implementation is available via `https://shenlirobot.github.io/pages/NeurIPS24.html`. The code is written in Julia and builds on the implementation by Tirinzoni and Degenne [63], where the transductive and weak-preference designs are solved using the Frank–Wolfe algorithm [24]. Their code is accessible at `https://github.com/AndreaTirinzoni/bandit-elimination`. Simulations and Bayesian inference for the DDM are implemented using the Julia package `SequentialSamplingModels.jl`, available at `https://itsdfish.github.io/SequentialSamplingModels.jl/dev/#SequentialSamplingModels.jl`.

For a query $x \in \mathcal{X}$, the estimators from Wagenmakers et al. [67] and Xiang Chiong et al. [73], analyzed in section 3.3 and benchmarked in section 5.2, require calculating $\mu^{-1}(p) := \log\left(p/\left(1-p\right)\right)$, where $\mu^{-1}(\cdot)$ is the logit function and $p := 1/n_x \cdot \sum_{i=1}^{n_x}\left(c_{x,s_{x,i}} + 1\right)/2$ represents the empirical mean of the human binary choices coded as 0 or 1. Since $p = 0$ or $p = 1$ makes this calculation undefined, we follow Wagenmakers et al. [67, the discussion below fig. 6] and approximate $p$ as $1 - 1/(2n_x)$ when $p = 1$ and $1/(2n_x)$ when $p = 0$.

## D.1  The "Sphere" Synthetic Problem for Evaluating Estimation Performance in section 5.1

We evaluate estimation performance using the "sphere" synthetic problem, a standard benchmark in linear bandit literature [20, 42, 61]. In this problem, the arm space $\mathcal{Z} \subset \{z \in \mathbb{R}^5 \colon \|z\|_2 = 1\}$ contains 10 randomly generated arms. To define the true preference vector $\theta^*$, we select the two arms $z$ and $z'$ that are closest in direction, i.e., $(z, z') \in \arg\max_{z,z' \in \mathcal{Z}} z^\top z'$, and set $\theta^* = z + 0.01(z' - z)$. In this way, $z$ is the best arm. The query space is $\mathcal{X} := \{z - z' \colon z \in \mathcal{Z}\}$.

## D.2  Processing the food-risk dataset with choices (-1 or 1) [57]

We accessed the food-risk dataset with choices (-1 or 1) [57] through Yang and Krajbich [76]'s repository (https://osf.io/d7s6c/). This dataset includes the choices and response times of 42 participants, each responding to between 60 and 200 queries. Each query compares two arms, with each arm containing two food items. By selecting an arm, participants had an equal chance of receiving either food item, hence the name "food risk" (or "food-gamble") task. Additionally, participants' eye movements were tracked during the experiment. Yang and Krajbich [76] modeled each participant's choices, response times, and eye movements using the attentional DDM [39], where the drift for each query is a linear combination of the participant's ratings of the four food items in the query, with the weights adjusting based on their eye movements. The ratings, $\in \{-10, -9, \ldots, 0, \ldots, 9, 10\}$, were collected before the participants interacted with the binary queries.

In our work, for each participant, we define each arm's feature vector as the participant's ratings of the two corresponding food items, augmented with second-order polynomials. We fit each participant's data to a difference-based EZ-diffusion model [8, 67] with a linear utility structure, as introduced in section 2. For each participant, using Bayesian inference with non-informative priors [16], we estimated the preference vector $\theta^* \in \mathbb{R}^5$, non-decision time $t_{\text{nondec}}$, and barrier $a$. Across participants, the barrier $a$ ranged from 0.715 to 2.467, with a mean of 1.437, and $t_{\text{nondec}}$ ranged from 0.206 to 1.917 seconds, with a mean of 0.746 seconds. This procedure generated one bandit instance per participant, with a preference vector $\theta^* \in \mathbb{R}^5$, an arm space $\mathcal{Z} \subset \mathbb{R}^5$ where $|\mathcal{Z}| \in [31, 95]$, and a query space $\mathcal{X} := \{z - z': z \in \mathcal{Z}\}$. Then, we used the fitted models to simulate human feedback for bandit experiments.

For each bandit instance, we benchmarked six GSE variations (introduced in section 5.2): $(\lambda_{\text{trans}}, \widehat{\theta}_{\text{CH,DT}})$, $(\lambda_{\text{trans}}, \widehat{\theta}_{\text{CH,RT}})$, $(\lambda_{\text{trans}}, \widehat{\theta}_{\text{CH}})$, $(\lambda_{\text{weak}}, \widehat{\theta}_{\text{CH}})$, $(\lambda_{\text{trans}}, \widehat{\theta}_{\text{CH,logit}})$, and $(\lambda_{\text{trans}}, \widehat{\theta}_{\text{CH,DT,logit}})$. For each GSE variation, we ran 300 repeated simulations under different random seeds, with human choices and response times sampled from the dEZDM with the identified parameters. Since each bandit instance contains a different number of arms, rather than tuning the elimination parameter $\eta$ in algorithm 1 for each instance, we set $\eta = 2$, following the convention in previous bandit research, e.g., Azizi et al. [3, section 3]. We manually tuned the buffer size $B_{\text{buff}}$ in algorithm 1 to 20, 30, or 50 seconds based on empirical performance, ensuring the budget was not exceeded in each phase. The full results are shown in fig. 5, with selected results highlighted in fig. 4a.

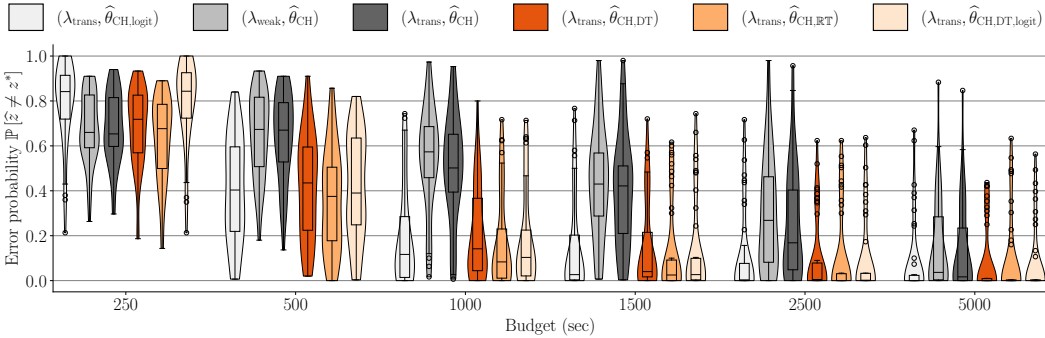

Figure 5: A violin plot overlaid with a box plot showing the best-arm identification error probability, $\mathbb{P}[\widehat{z} \neq z^*]$, as a function of budget for each GSE variation, simulated using the food-risk dataset with choices (-1 or 1) [57], as described in appendix D.2. The box plots follow the convention of the matplotlib Python package. For each GSE variation and budget, the horizontal line in the middle of the box represents the median of the error probabilities across all bandit instances. Each error probability is averaged over 300 repeated simulations under different random seeds. The box's upper and lower borders represent the third and first quartiles, respectively, with whiskers extending to the farthest points within $1.5\times$ the interquartile range. Flier points indicate outliers beyond the whiskers.

### D.3 Processing the snack dataset with choices (yes or no) [16]

We accessed the snack dataset with choices (yes or no) [16] through the supplementary material provided by Alós-Ferrer et al. [2] at `https://www.journals.uchicago.edu/doi/abs/10.1086/713732`. This dataset consists of training and testing data. The training data was collected from a "YN" task, where 31 participants provided binary feedback ("Yes" or "No") and response times for queries comparing each of the 17 snack items to a fixed reference snack, with each query repeated 10 times. The reference snack, assigned a utility of 0, remained fixed throughout the experiment. The testing data was collected using a two-alternative forced-choice task, where participants provided binary choices and response times for queries comparing two snack items, with each query repeated once. Clithero [16] fit a difference-based EZ-diffusion model [8, 67] to the training data using Bayesian inference with non-informative priors, without imposing a linear utility structure, and tested the model using the testing data.

In our work, we fit each participant's training data to a difference-based EZ-diffusion model with a linear utility structure, as described in section 2, and used the fitted model to simulate human feedback for bandit experiments. We preprocessed the data by removing outliers, following Clithero [16, footnote 22], excluding trials with response times below 200 ms or greater than five standard deviations above the mean. After cleaning, the number of trials per participant ranged from 167 to 170. Since the dataset does not provide feature vectors for the 17 non-reference snack items, we used one-hot encoding to represent each snack item as a feature vector in $\mathbb{R}^{17}$. This allowed us to construct a bandit instance for each participant with a preference vector $\theta^* \in \mathbb{R}^{17}$, an arm space $\mathcal{Z} \subset \mathbb{R}^{17}$ with $|\mathcal{Z}| = 17$, and a query space $\mathcal{X} := \{z - \mathbf{0} : z \in \mathcal{Z}\}$ to represent comparisons with the reference snack. We applied Bayesian inference with non-informative priors [16] to estimate each participant's preference vector $\theta^*$, non-decision time $t_{\text{nondec}}$, and barrier $a$. Across participants, the barrier $a$ ranged from $0.759$ to $1.399$, with a mean of $1.1$, and $t_{\text{nondec}}$ ranged from $0.139$ to $0.485$ seconds, with a mean of $0.367$ seconds.

For each of the six GSE variations (introduced in section 5.2): $(\lambda_{\text{trans}}, \widehat{\theta}_{\text{CH,DT}})$, $(\lambda_{\text{trans}}, \widehat{\theta}_{\text{CH,}\mathbb{RT}})$, $(\lambda_{\text{trans}}, \widehat{\theta}_{\text{CH}})$, $(\lambda_{\text{weak}}, \widehat{\theta}_{\text{CH}})$, $(\lambda_{\text{trans}}, \widehat{\theta}_{\text{CH,logit}})$, and $(\lambda_{\text{trans}}, \widehat{\theta}_{\text{CH,DT,logit}})$, we tuned the elimination parameter $\eta$ in algorithm 1 using the following procedure: We considered $\eta \in \{2, 3, 4, 5, 6, 7, 8, 9\}$, resulting in the number of phases $:= \lceil \log_\eta |\mathcal{Z}| \rceil = \lceil \log_\eta(17) \rceil$ (line 4 of algorithm 1) being $\{5, 3, 3, 2, 2, 2, 2, 2\}$, respectively. We excluded $\eta > \lceil 17/2 \rceil = 9$, as those cases also result in 2 phases, the same as $\eta \in \{5, 6, 7, 8, 9\}$. Then, for each $\eta$, for each of the 31 bandit instances, and for each budget $\in \{50, 75, 100, 125, 150, 200, 250, 300\}$ seconds, we ran 50 repeated simulations per GSE variation under different random seeds, sampling human feedback from the fitted dEZDM. We then aggregated the results into a single best-arm identification error probability for each GSE variation, $\eta$, bandit instance, and budget. These error probabilities were compiled into violin and box plots, as shown in fig. 6.

For each GSE variation, we selected the $\eta$ that minimized the median error probability, as shown in the box plots in fig. 6. If multiple $\eta$ values yielded the same median, we used the third quartile, and if necessary, the first quartile, to break ties. Based on this approach, we selected: $\eta = 6$ for $(\lambda_{\text{trans}}, \widehat{\theta}_{\text{CH,DT}})$, $\eta = 6$ for $(\lambda_{\text{trans}}, \widehat{\theta}_{\text{CH,}\mathbb{RT}})$, $\eta = 9$ for $(\lambda_{\text{trans}}, \widehat{\theta}_{\text{CH}})$, $\eta = 9$ for $(\lambda_{\text{weak}}, \widehat{\theta}_{\text{CH}})$, $\eta = 9$ for $(\lambda_{\text{trans}}, \widehat{\theta}_{\text{CH,logit}})$, and $\eta = 5$ for $(\lambda_{\text{trans}}, \widehat{\theta}_{\text{CH,DT,logit}})$.

After tuning $\eta$, we manually set the buffer size $B_{\text{buff}}$ in algorithm 1 to 10 seconds based on empirical results, ensuring the budget was not exceeded in any phase. We then benchmarked each GSE variation on all 31 bandit instances using its own manually tuned $\eta$ and $B_{\text{buff}}$. Each variation was evaluated over 300 repeated simulations with different random seeds, where human choices and response times were sampled from the dEZDM with the identified parameters. The full results are shown in fig. 7, with selected results presented in fig. 4b.

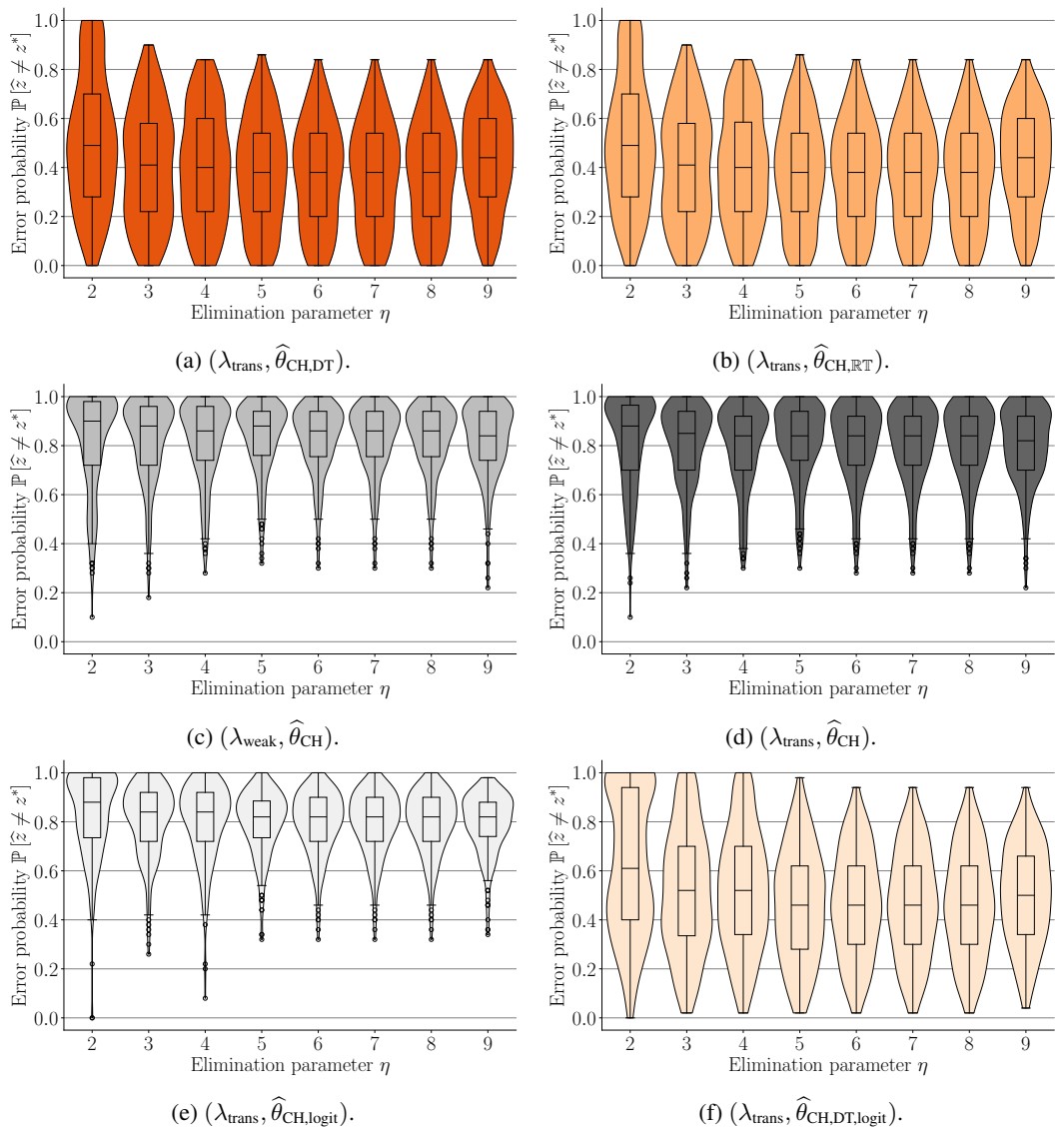

Figure 6: Violin plots overlaid with box plots, used for tuning the elimination parameter $\eta$ in algorithm 1 for each GSE variation, simulated based on the snack dataset with choices (yes or no) [16], as discussed in appendix D.3. Each plot shows the best-arm identification error probability, $\mathbb{P}\left[\widehat{z} \neq z^*\right]$, as a function of $\eta$. The box plots follow the convention of the `matplotlib` Python package. The horizontal line in each box represents the median of the error probabilities across all bandit instances and budgets. Each error probability is averaged over 50 repeated simulations under different random seeds. The top and bottom borders of the box represent the third and first quartiles, respectively, while the whiskers extend to the farthest points within $1.5\times$ the interquartile range. Flier points are the outliers past the end of the whiskers.

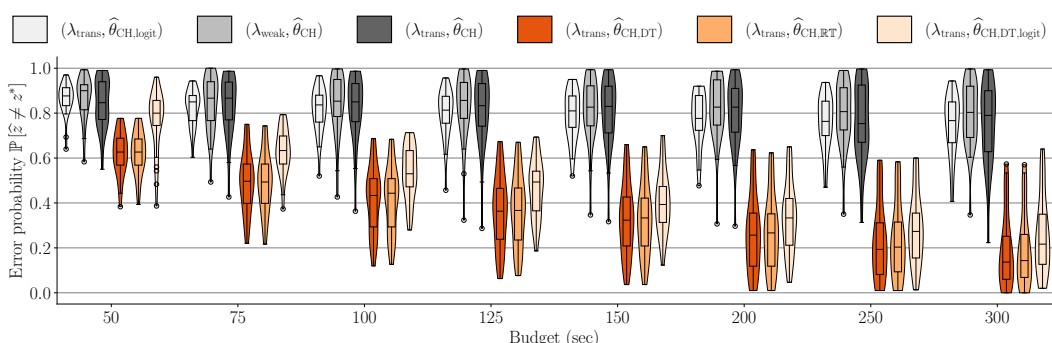

Figure 7: A violin plot overlaid with a box plot showing the best-arm identification error probability, $\mathbb{P}\left[\widehat{z} \neq z^*\right]$, as a function of budget for each GSE variation, simulated using the snack dataset with choices (yes or no) [16], as described in appendix D.3. The box plots follow the convention of the `matplotlib` Python package. For each GSE variation and budget, the horizontal line in the middle of the box represents the median of the error probabilities across all bandit instances. Each error probability is averaged over 300 repeated simulations under different random seeds. The box's upper and lower borders represent the third and first quartiles, respectively, with whiskers extending to the farthest points within $1.5\times$ the interquartile range. Flier points indicate outliers beyond the whiskers.

### D.4 Processing the snack dataset with choices (-1 or 1) [39]

We accessed the snack dataset with choices (-1 or 1) [39] via Fudenberg et al. [27]'s replication package at `https://www.aeaweb.org/articles?id=10.1257/aer.20150742`. This dataset contains choices and response times from 39 participants, each responding to between 49 and 100 queries comparing two snack items. Participants' eye movements were tracked during the experiment. Krajbich et al. [39] modeled each participant's choices, response times, and eye movements using the attentional DDM, where the drift for each query is a linear combination of the participant's ratings of both snack items in the query, with the weights influenced by their eye movements. The ratings, $\in \{-10, -9, \ldots, 0, \ldots, 9, 10\}$, were collected before participants interacted with the binary queries.

In our work, to avoid creating trivial bandit problems by encoding snack items as 1-dimensional vectors (as done in appendix D.2), we defined the feature vector for each snack item with a participant rating $r_z \in \{-10, -9, \ldots, 0, \ldots, 9, 10\}$ as a one-hot vector in $\mathbb{R}^{21}$, where the $(r_z + 11)$-th element is 1 and the rest are 0. The preference vector $\theta^*$ is structured as $\beta^* \cdot [-10, -9, \ldots, 0, \ldots, 9, 10]^\top \in \mathbb{R}^{21}$, where $\beta^*$ is participant-specific and unknown to the learner. This ensures that, for each arm $z$, the participant's utility is $u_z := z^\top \theta^* = r_z \beta^*$. In this way, each participant's data generated a bandit instance with a preference vector $\theta^* \in \mathbb{R}^{21}$, a set of arms $\mathcal{Z} \subset \mathbb{R}^{21}$ with $|\mathcal{Z}| = 21$, and a query space $\mathcal{X} := \{z - z' : z \in \mathcal{Z}\}$.

We fit each participant's data to a difference-based EZ-diffusion model [8, 67] using the linear utility structure described above. For each participant, using Bayesian inference with non-informative priors [16], we estimated the preference vector $\theta^*$ (or equivalently, the parameter $\beta^*$), non-decision time $t_{\text{nondec}}$, and barrier $a$. Across participants, the barrier $a$ ranged from 0.75 to 2.192 with a mean of 1.335, and $t_{\text{nondec}}$ ranged from 0.387 to 1.22 seconds with a mean of 0.641 seconds. We then used these fitted models to simulate human feedback for bandit experiments, assuming the learner did not know the underlying structure $\theta^* = \beta^* \cdot [-10, -9, \ldots, 0, \ldots, 9, 10]^\top$.

For each of the following GSE variations (introduced in section 5.2): $(\lambda_{\text{trans}}, \widehat{\theta}_{\text{CH,DT}})$, $(\lambda_{\text{trans}}, \widehat{\theta}_{\text{CH},\mathbb{RT}})$, $(\lambda_{\text{trans}}, \widehat{\theta}_{\text{CH}})$, $(\lambda_{\text{weak}}, \widehat{\theta}_{\text{CH}})$, $(\lambda_{\text{trans}}, \widehat{\theta}_{\text{CH,logit}})$, and $(\lambda_{\text{trans}}, \widehat{\theta}_{\text{CH,DT,logit}})$, we tuned the elimination parameter $\eta$ in algorithm 1 using the following procedure: We considered $\eta \in \{2, 3, 4, 5, 6, 7, 8, 9, 10, 11\}$, which resulted in the number of phases $:= \lceil \log_\eta |\mathcal{Z}| \rceil = \lceil \log_\eta(17) \rceil$ (line 4 of algorithm 1) being $\{5, 3, 3, 2, 2, 2, 2, 2, 2, 2\}$, respectively. We excluded cases where $\eta > \lceil 21/2 \rceil = 11$, as these result in 2 phases, identical to when $\eta \in \{5, 6, 7, 8, 9, 10, 11\}$. Then, for each $\eta$, for each of the 39 bandit instances, and for each budget $\in \{150, 200, 250, 300, 350, 400, 450, 500\}$ seconds, we ran 50 repeated simulations per GSE variation under different random seeds, sampling human feedback from the fitted dEZDM. We then aggregated the results into a single best-arm identification error probability for each GSE variation, $\eta$, bandit instance, and budget. These error probabilities were compiled into violin and box plots, as shown in fig. 8.

For each GSE variation, we selected the $\eta$ that minimized the median error probability, as shown in the box plots in fig. 8. If multiple $\eta$ values yielded the same median, we used the third quartile, and if necessary, the first quartile, to break ties. Based on this approach, we selected: $\eta = 4$ for $(\lambda_{\text{trans}}, \widehat{\theta}_{\text{CH,DT}})$, $\eta = 4$ for $(\lambda_{\text{trans}}, \widehat{\theta}_{\text{CH},\mathbb{RT}})$, $\eta = 4$ for $(\lambda_{\text{trans}}, \widehat{\theta}_{\text{CH}})$, $\eta = 2$ for $(\lambda_{\text{weak}}, \widehat{\theta}_{\text{CH}})$, $\eta = 5$ for $(\lambda_{\text{trans}}, \widehat{\theta}_{\text{CH,logit}})$, and $\eta = 5$ for $(\lambda_{\text{trans}}, \widehat{\theta}_{\text{CH,DT,logit}})$.

After tuning $\eta$, we manually set the buffer size $B_{\text{buff}}$ in algorithm 1 to 20 seconds based on empirical results, ensuring the budget was not exceeded in any phase. We then benchmarked each GSE variation on all 39 bandit instances using its own manually tuned $\eta$. Each variation was evaluated over 300 repeated simulations with different random seeds, where human choices and response times were sampled from the dEZDM with the identified parameters. The full results are shown in fig. 9, with selected results presented in fig. 4c.

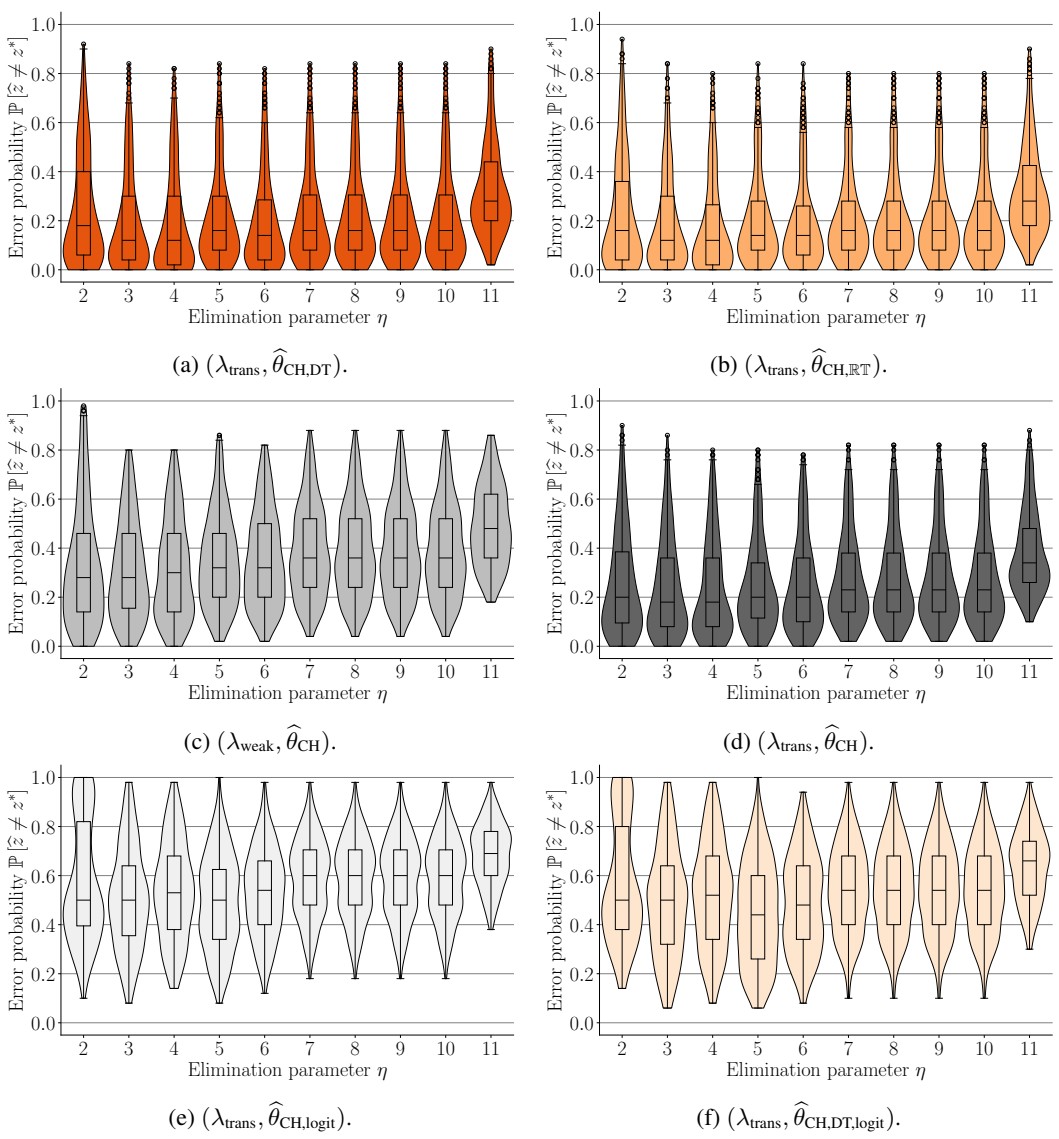

Figure 8: Violin plots overlaid with box plots, used for tuning the elimination parameter $\eta$ in algorithm 1 for each GSE variation, simulated based on the snack dataset with choices (-1 or 1) [39], as discussed in appendix D.4. Each plot shows the best-arm identification error probability, $\mathbb{P}\left[\widehat{z} \neq z^*\right]$, as a function of $\eta$. The box plots follow the convention of the matplotlib Python package. The horizontal line in each box represents the median of the error probabilities across all bandit instances and budgets. Each error probability is averaged over 50 repeated simulations under different random seeds. The top and bottom borders of the box represent the third and first quartiles, respectively, while the whiskers extend to the farthest points within $1.5\times$ the interquartile range. Flier points are the outliers past the end of the whiskers.

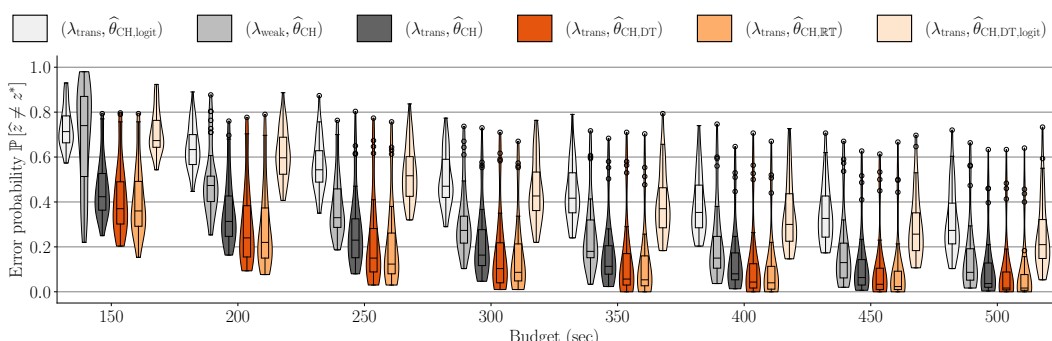

Figure 9: A violin plot overlaid with a box plot showing the best-arm identification error probability, $\mathbb{P}\left[\widehat{z} \neq z^*\right]$, as a function of budget for each GSE variation, simulated using the snack dataset with choices (-1 or 1) [39], as described in appendix D.4. The box plots follow the convention of the `matplotlib` Python package. For each GSE variation and budget, the horizontal line in the middle of the box represents the median of the error probabilities across all bandit instances. Each error probability is averaged over 300 repeated simulations under different random seeds. The box's upper and lower borders represent the third and first quartiles, respectively, with whiskers extending to the farthest points within $1.5\times$ the interquartile range. Flier points indicate outliers beyond the whiskers.

