# OpenReview forum: "Enhancing Preference-based Linear Bandits via Human Response Time"
_NeurIPS.cc/2024/Conference — NeurIPS 2024 oral_

### Official Review · Reviewer_64dM · 2024-07-11

**Soundness:** 3
**Presentation:** 3
**Contribution:** 3
**Rating:** 7
**Confidence:** 3

**Summary:**

The paper explores the interactive preference learning problem.
Traditional binary choice feedback is limited in conveying preference strength.
The authors propose leveraging human response time, which inversely correlates with preference strength, as additional information.
They adopt the difference-based Drift-Diffusion Model with linear human utility functions and propose a computationally efficient linear utility estimator that incorporates human response times.
Discussions about the proposed estimator and traditional estimators relying on the logistic regression are provided.
The authors also integrate the proposed estimator into the Generalized Successive Elimination(GME) algorithm for the fixed-budget best arm identification problem.
The proposed method is evaluated on both synthetic and real-world datasets, demonstrating the effectiveness of incorporating human response times for essay queries.

**Strengths:**

- The idea of incorporating human response times is novel and interesting. The authors provide a clear motivation to incorporate human response times, give illustrative examples to illustrate the benefits and explain their methods in detail.
- The paper is well-written and easy to follow. The background, methods, and experiments are well-organized and well-explained.
- The authors clearly discuss the effects of parameters $x^T\theta^{*}$ and $a$, and compare the estimator with and without response times, which provides a good understanding of the proposed method.
- The authors conduct extensive experiments on both synthetic and real-world datasets, and provide a detailed analysis of the results about the estimation performance and identification performance. The methods to pre-process the datasets and determine the parameters are also well explained.

**Weaknesses:**

- Although the authors provide the asymptotic analysis for their estimators, they do not provide a theoretical analysis for the proposed algorithm.
- As mentioned in the paper, when the barrier $a$ is small, incorporating response times may not improve estimation performance. Since the parameter $a$ is also unknown, like $\theta^{*}$, learners may not know whether to incorporate response times. It would be beneficial if the authors could have a discussion about the practical choice of the estimator under these conditions.

**Questions:**

Please refer to the weaknesses mentioned above.

**Limitations:**

Limitations are adequately discussed in the paper.

---

> ### Author Rebuttal · Authors · 2024-08-07
>
> Thank you for your detailed and thoughtful review, and for recognizing the novelty and clarity of our work.
>
> ## Weakness: algorithm analysis
> In our paper draft, we provide asymptotic theoretical results to show the following intuition on why response time can improve the learning performance:
>
> *Combining response times  and choices, information from easy queries can be extracted more efficiently. This is not possible using choices only.*
>
> We added non-asymptotic error probability on estimating reward $x^{\top}\theta^*/a$ for every query $x$ using both methods, i.e. combining response times with choices and using choices only. Similar intuition can be confirmed as is shown in the `Author Rebuttal' section.
>
> We leave for future work to provide non-asymptotic error probability for the entire algorithm (Algoithm 1 in the paper).
>
> ## Weakness: practical usage of response time when $a$ is unknown
> One straightforward approach is to always incorporate response times:
> 1. Based on our empirical results, it seems that incorporating response times, if not improving the performance, rarely degrades performance.
> 2. Our theoretical and empirical results indicate that when queries are very difficult for humans to answer, incorporating response times may be less beneficial and could slightly decrease performance. However, in these scenarios, humans typically don’t have strong preferences, so they might be more tolerant of the minor performance impact caused by using response times.
>
> Alternatively, one can first incorporate response times to filter out many suboptimal options, and then use choice-only estimation for further learning. In the first stage, both good and bad arms exist. In this case, many queries, composed of one good arm and one bad arm, are easy and response times help extract more information from those easy queries. In the second stage, most arms will be similarly good. In this case, the queries are harder and choice-only estimation is sufficient to identify the best arm. Determining the optimal point to switch from using response times to relying solely on choices is an area for future research.

---

> > ### Comment · Reviewer_64dM · 2024-08-10
> >
> > The reviewer thanks the authors for addressing their concerns. The reviewer will raise the score accordingly.

---

### Official Review · Reviewer_a7WX · 2024-07-12

**Soundness:** 3
**Presentation:** 3
**Contribution:** 3
**Rating:** 7
**Confidence:** 3

**Summary:**

The paper studies linear bandit preference learning, where binary preference data have been augmented with response times. A joint model for choice and response time falls from setting the linear preference model as the drift parameter in a drift-diffusion model. Experiments and theoretical analysis show that including response time in the model increases the value of "easy" responses, and thus improves bandit performance.

**Strengths:**

The paper is well-written. Including response time in a linear bandit preference model is novel to my knowledge, and certainly useful. The development of the method is clear, and the method itself is well-motivated and seems computationally reasonable and useful. The theoretical analysis is helpful, and I really appreciated how the paper uses that analysis to draw insights into the source of improvement from including response times in the model. The experiments were well-designed, and included the necessary baseline of choice-only. The analysis of the experiments provides helpful guidance for the situations in which the method does not outperform baselines.

**Weaknesses:**

The paper has one significant weakness: all of the experiments use simulated response times, simulated from the model developed in the paper. We thus don't get a sense for the "real-world" performance of the method, where response times will not adhere precisely to the DDM model. I agree with the paper that response time is easy to collect alongside binary preferences. Surely there are some datasets available that include actual human response times? Real human response times are the big missing piece of the experimental evaluation.

**Questions:**

On the topic of real human response time data, should we expect the model to be robust to lapses, or would some sort of data pre-processing be necessary to remove those?

**Limitations:**

Addressed.

---

> ### Author Rebuttal · Authors · 2024-08-07
>
> Thank you for your positive and constructive review; we appreciate your feedback and recognition of our work's novelty and usefulness.
>
> ## Weakness: real-world empirical result
> We acknowledge the importance of using real-world data for evaluation. Below is the rationale behind our simulator-based study and additional results using a different human dataset, as detailed in the `Author Rebuttal` section.
>
> Firstly, our bandit algorithm is an online algorithm, requiring us to sample response times of different queries online adaptively. Existing offline datasets are not collected adaptively, and is hard to be used for evaluating our algorithm. Although, there exist offline policy evaluation methods[2] that evaluate online algorithms with offline data, they require extensive data from a single user. Unfortunately, we have not been able to find such large-scaled datasets for response times.
>
> Secondly, online collecting response time data can suffer from outliers due to human inattention or anticipation [1]. Successful studies may require integrating data cleansing techniques [1] into online algorithms, which we consider as a separate contribution in future work.
>
> Therefore, in our work, we adopt a third approach: training a simulator from real-world data and then using the simulator to evaluate algorithms. This approach assumes that the EZ-DDM is the ground truth model, which is a reasonable assumption given the empirical support for this model [3]. We believe this approach justifies our insights—using response times makes easy queries more useful—while leaving a full user study for future work.
>
> We have included a new simulation-based study using another dataset of human choices and response times in the `Author Rebuttal` section. This study confirms that incorporating response times improves best-arm identification performance.
>
> - [[1] Myers et al. 2022](https://www.frontiersin.org/journals/psychology/articles/10.3389/fpsyg.2022.1039172/full)
> - [[2] Li et al. 2010](https://dl.acm.org/doi/abs/10.1145/1772690.1772758?casa_token=RuMgk8seZScAAAAA:YyJgdyVIfKDBquZ8uuDO-RAg3kK3vVmu-3Drco_J8CCUxYJXGgg2TCUepSwV6UvWqU4hYbpNwDTbVg)
> - [[3] Wagenmakers et al. 2007](https://link.springer.com/article/10.3758/bf03194023)
>
> ## Question: about lapses and data processing
> In general, there are two types of lapses: 1) lapses at the very beginning of the decision process; 2) lapses, or distractions, in the middle of the decision process. The first type of lapse can be interpreted as the non-decision time in the EZ-DDM model. The appendix includes our discussions about the issue of unknown non-decision times. Future work could involve estimating non-decision times from data.
>
> The second type of lapse occurs when humans lose attention or get distracted during the decision process, resulting in very long response times. Alternatively, humans might anticipate the current trial based on previous trials, leading to very short response times [2]. To handle such outliers in real-world datasets, a common procedure is to define cut-off thresholds to eliminate very short and very long response times [2]. Additionally, human attention can be monitored using eye-tracking devices. There is a line of psychological literature [1] that tracks human eye gazes during decision-making and incorporates human visual attention within the DDM framework.
>
> Another important procedure for handling real-world response time data is to test whether DDM is an appropriate model for a given dataset. There is literature on statistically testing whether observed response time data is generated by DDM [3][4].
>
> - [[1] Krajbich 2019](https://www.sciencedirect.com/science/article/abs/pii/S2352250X18301866)
> - [[2] Myers et al. 2022](https://www.frontiersin.org/journals/psychology/articles/10.3389/fpsyg.2022.1039172/full)
> - [[3] Alós-Ferrer et al. 2021](https://www.journals.uchicago.edu/doi/full/10.1086/713732)
> - [[4] Fudenberg et al. 2020](https://www.pnas.org/doi/abs/10.1073/pnas.2011446117)

---

> > ### Comment · Reviewer_a7WX · 2024-08-13
> >
> > Thank you for the additional experiments and analysis. I will raise my score as a result and hope that the paper is accepted. There was some related work just published at UAI that may be worth mentioning: Shvartsman et al. "Response time improves Gaussian process models for perception and preferences", https://openreview.net/forum?id=oUZ5JweNRc . It's essentially hooking a DDM model up into a Gaussian process bandit. Similar in concept though the mechanics and application area are of course quite different.

---

### Official Review · Reviewer_qjfL · 2024-07-13

**Soundness:** 4
**Presentation:** 3
**Contribution:** 4
**Rating:** 8
**Confidence:** 5

**Summary:**

The submission proposes to use response times to obtain additional information from participants in preference learning settings. They apply a variant of the Drift-Diffusion Model, a popular model of human decision making from psychology, and combine it with an algorithm applicable to linear bandits. In asymptotic analysis and application to real data, they demonstrate the benefits of taking advantage of response times, especially for queries with large value differences.

**Strengths:**

I think this is a very sensible approach to improving preference learning, considering the fact that response times can be available "for free" from existing preference learning paradigms. I also think the application of a simplified variant of the DDM (EZ-Diffusion) as a way to get an analytic handle on things is also a good idea, and I think the results are compelling. It's a good paper, and I enjoyed reading it.

**Weaknesses:**

* I think the paper slightly misrepresents the actual model it's using. Unless I'm missing something, the DDM is usually defined as the first passage time of a two-boundary Wiener process, and the likelihood of a RT under that model involves solving an SDE which has no closed-form solution. However, moments of the DDM RT distribution are available, which is what enables approaches that use them to approximately solve for parameters (this includes the E-Z Diffusion model of Wagenmakers et al., the current contribution (as far as I can tell), as well as the related work of Shvartsman et al. 2023 (arxiv:2306.06296). I think using E-Z diffusion type approaches is great, I just think the paper shouldn't represent them as the full DDM. Also note that the E-Z diffusion line of work provides for closed-form estimation of nondecision time (by estimating drift (i.e. value difference) based on response time variance, and then backing into the response time mean from there -- this might help address the issue of unknown nondecision time identified by the authors.
* Related to the above, moving away from the simplifying assumption of E-Z diffusion would introduce the concern of other unknown DDM parameters such as drift variability, nonsymmetrical initial conditions, etc.

**Questions:**

* Why should choice-only would ever do better asymptotically. L186 makes this claim w.r.t. section 3.2 but I don't see it in the figure -- the orange lines (dashed or solid) seem always above the gray lines.
* For figure 2, if possible it should be moved below section 3.2., so that it can be interpreted after the asymptotic min-weight of both estimators has been discussed. This is also the part of the submission which is most confusing -- mental calisthenics are needed to map from asymptotic min-weight to the variance of theta estimates and therefore to the influence of observations at various value differences. Another editing / clarification pass could be helpful.

Additional notes:
* All figure axis tick labels are very tiny, should ideally be larger. In addition, adding color legends in fig 4 (outside the caption) would help with readability.

**Limitations:**

Discussed and addressed, even with additional analyses (and as noted above, the limitation regarding nondecision time might be possible to work around, though I could be missing something there).

---

> ### Author Rebuttal · Authors · 2024-08-07
>
> Thank you for your detailed and positive review. We appreciate your constructive feedback and are glad you enjoyed the paper. We will reorganize the theoretical sections and figure presentation as suggested.
>
> ## Weakness: about EZ-DDM vs DDM and EZ-DDM's assumptions
> Thank you for pointing out the distinction between DDM and EZ-DDM [1]. We will ensure that our language in the paper accurately reflects this difference. Our work indeed adopts the assumptions of EZ-DDM, including deterministic starting points (zero-valued), drift, and non-decision time. Lifting these assumptions within the bandit framework could be a fruitful direction for future research.
>
> As mentioned, our work assumes known non-decision time for simplicity. We appreciate your reference to EZ-DDM's procedure for estimating non-decision time (Eq.9 of [1]), and we agree that integrating this method with our approach is a promising avenue for future work. One reason we have not yet incorporated this method is that the estimation procedures in [1] and [2] treat each query’s non-decision time separately. In contrast, our work, following [3], assumes a common non-decision time across all queries. A potential future direction is to aggregate data across queries via the common non-decision time, similar to how we aggregate data in our current draft via the linear utility structure.
>
> To compare our estimator with those in [1], consider the estimation of the drift $u_x$ for a query $x$. Our choice-and-response-time estimator (Eq. 4 in our draft) becomes the ratio of the expected choice and the expected response time. In contrast, [1]'s estimator (Eq. 5 in [1]) is based solely on the choices' log-odds, $\log\left(\mathbb{P}(c_x=1)/(1-\mathbb{P}(c_x=1))\right)$. As our non-asymptotic analysis in `Author Rebuttal` indicates, our estimator can perform better for easy queries, while the choice-based estimator may be more effective for hard queries. When the utilities are parameterized linearly, our choice-and-response-time estimator is Eq. 4 in our paper draft, whereas [1]'s estimator becomes Eq. 5 in our paper draft. Our asymptotic analysis in Theorems 3.1 and 3.2 again highlights that using response times can be beneficial for easy queries.
>
> - [1] [Wagenmakers et al. 2007](https://link.springer.com/article/10.3758/BF03194023)
> - [2] [Berlinghieri et al. 2023](https://www.science.org/doi/10.1126/sciadv.adf1665)
> - [3] [Clithero (2018)](https://www.sciencedirect.com/science/article/abs/pii/S0167268118300398})
>
> ## Question: why choice-only could perform better asymptotically?
> Given some arm $y\in\mathcal{Z}$, we use asymptotic variance to measure how much the estimated utility $y^T\widehat{\theta}$ varies around the true utility value $y^T\theta^*$ when the sample size is very large. Our goal is to compare the asymptotic variance of the choice-response-time estimator to that of the choice-only estimator.
>
> Both estimators' asymptotic variances depend on how much information the estimator retain from the data, i.e. human responses to queries $x\in \mathcal{X}\_{sample}$.
> The choice-response-time estimator and the choice-only estimator retain different aspects of information from $x$. Intuitively, choices retain the "sign" of the utility difference $x^T\theta^*$, while response times retain the preference strength. The "amount" of information they retained is formally presented as the weights $\mathcal{M}\_{CH,DT}$ and $m_{CH}$ in Theorems 3.1 and 3.2, respectively. Higher values of these terms indicate more retained information, leading to lower variance and better estimation.
>
> We compare the weights $\mathcal{M}\_{CH,DT}$ and $m_{CH}$ in Theorems 3.1 and 3.2. The choice-only estimator assigns each query $x$ a weight $m_{CH}(x^T\theta^*)$, represented by the gray curve in Figure 2. In contrast, the choice-response-time estimator assigns all queries the same weight $\mathcal{M}\_{CH,DT}\coloneqq\min_{x}m_{CH,DT}(x^T\theta^*)$, with $m_{CH,DT}(x^T\theta^*)$ plotted as the orange curve in Figure 2.
>
> While the orange curve is consistently higher than the gray curve, indicating that $m_{CH,DT}(x^T\theta^*)>m_{CH}(x^T\theta^*)$ for each query $x$, the choice-response-time estimator's weight is $\mathcal{M}\_{CH,DT}$, not $m_{CH,DT}(x^T\theta^*)$. Consequently, $\mathcal{M}\_{CH,DT}$ may be larger or smaller than $m_{CH}(x^T\theta^*)$ depending on the queries in the data.
> For instance, if the data contains both hard queries where $x^T\theta^*\in[-1,1]$ and one easy query where $x^T\theta^*=4$, the choice-response-time estimator will have small weights for all queries due to the "min" in the definition of $\mathcal{M}\_{CH,DT}$, while the choice-only estimator will have large weights for hard queries even if the easy query exists. In this scenario, the choice-only estimator may perform better. Conversely, if the data only contain easy queries, the choice-response-time estimator will have a larger weight, making it superior.
>
> Finally, we would like to note that the "min" in the definition of $\mathcal{M}\_{CH,DT}$ is a result of our proof techniques. It is possible to obtain tighter theoretical analysis, and we leave it for future work.

---

> > ### Comment · Reviewer_qjfL · 2024-08-13
> >
> > I appreciate the clarifications regarding EZ-DDM and the asymptotic behavior of the choice-only strategy. I recommend that the authors try to make room for both clarifications (the distinction between the paper's approach and EZ-DDM's estimators, and the limiting case where response times don't help) in the main text or at least the supplement, alongside the other minor reorganizations.
> >
> > I'm happy to see that the other reviewers agree that this is a good paper. I will add more and say that it's the most straightforwardly applicable approach to using response times in preference learning that I've seen (without requiring MCMC sampling, likelihood approximations, or funky numerics), and the only one with any sort of non-vacuous theoretical guarantees. As such, it's the most likely to have impact more broadly (e.g. on current "hot" areas like RLHF). I have raised my rating by a point accordingly, and hope to see it at the conference.

---

### Official Review · Reviewer_f1Vk · 2024-07-17

**Soundness:** 3
**Presentation:** 4
**Contribution:** 3
**Rating:** 7
**Confidence:** 3

**Summary:**

This paper studies whether leveraging human response times can lead to better performance in bandit learning from preference feedback. More specifically, the paper integrates the drift-diffusion model (DDM) from psychology into the best-arm identification problem in linear bandits. Given a fixed interaction time, the goal is to utilize response times alongside binary choices so as to maximize the probability of recommending the optimal arm.

The paper introduces an estimator of the preference/reward vector using both binary responses and response times via linear regression. This estimator can be incorporated into bandit algorithms. Asymptotic normality results and three simulations indicate that this new estimator leveraging response times can make easier queries more useful, in comparison with traditional estimators that only use binary responses.

**Strengths:**

- The studied problem seems novel, interesting, and relevant to the community. To my knowledge, DDMs have not been (widely) explored in bandits and reinforcement learning, although I am not very familiar with the DDM literature in psychology and neuroeconomics. This model may be of particular interests to researchers studying dueling bandits and RLHF.

- The model leads to a simple and clean estimator of human preferences $\theta^*$, which uses both binary preference feedback and response times. To my understanding, this estimator can be integrated into various bandit algorithms, not limited to ones for best-arm identification.

- The paper presents both theoretical and empirical evidence on the role of response times, and discusses the intuition behind when/why they can be useful.

- The paper is well-organized and well-written. It is a joy to read.

**Weaknesses:**

- It would be helpful to see more discussion on related work, especially on DDMs and race models. For example, what evidence has been given in the psychology literature that DDMs can explain the human decision-making process? What are the typical objectives in papers that study DDMs, and how are they different from the best-arm identification problem in this work? Have DDMs been considered in bandits and reinforcement learning?

- On the theoretical side, there are no non-asymptotic results regarding the performance of the bandit algorithm, such as bounds on the error probability.

- On the empirical side, only one dataset contains the response times of the participants. The last two experiments simulate response times according to the DDM -- it is reasonable to expect that they are then useful to estimating $\theta^*$. The algorithm also has a hyperparameter that requires tuning.

**Questions:**

See above for questions on related work.

I am also curious -- the estimator that uses only binary responses essentially assumes the Bradley-Terry model, if my understanding is correct. Looking at the logistic sigmoid function, I can see that the when $x^\top \theta^*$ is away from 0, the curve becomes flat and therefore not much information can be gained here. Have you considered other noise models that do not use a link function like the logistic sigmoid function? Would you expect similar behavior?

**Limitations:**

Limitations have been discussed.

---

> ### Author Rebuttal · Authors · 2024-08-07
>
> Thank you for your detailed and thoughtful review. We appreciate your recognition of the novelty of our work and positive feedback on the writing. Here are our responses to your concerns and questions:
>
> ## Weakness: non-asymptotic result
> In our paper draft, we provide asymptotic theoretical results to show the following intuition on why response time can improve the learning performance:
>
> *Combining response times  and choices, information from easy queries can be extracted more efficiently. This is not possible using choices only.*
>
> We added non-asymptotic error probability on estimating reward $x^{\top}\theta^*/a$ for every query $x$ using both methods, i.e. combining response times with choices and using choices only. Similar intuition can be confirmed as is shown in the `Author Rebuttal' section.
>
> We leave for future work to provide non-asymptotic error probability for the entire algorithm (Algoithm 1 in the paper).
>
> ## Weakness: real-world empirical result
> We included a new simulation study based on another dataset of human choices and response times in the `Author Rebuttal` section. This study shows that incorporating response times improves best-arm identification performance.
>
> ## Weakness+question: background of DDM models
> We plan to include the following summary of DDM literature in the appendix:
>
> #### 1. Literature on modeling choices and response times
> Bounded accumulation models (BAMs) capture the human decision-making process with an accumulator and a stopping rule. For binary choices, DDM [1] models the human's speed-accuracy trade-off with one accumulator, fixed barriers, random starting points, drift, and non-decision times. Our paper adopts EZ-DDM [3], a simplified version with deterministic parameters.
> DDM with time-decaying barriers theoretically connects to human Bayesian RL models [5].
> DDMs also characterize human attention during decision-making, by modeling choices, response times, and eye gazes on options or attributes [7].
> Race models [4] extends to queries with more than two options by assuming an accumulator for each option and stopping when any accumulator reaches its threshold.
> Neurophysiological evidence supports BAMs. EEG recordings show neurons exhibit accumulation processes and decision thresholds [2][6].
>
>
> #### 2. Literature on using response times (survey [10])
> Response times improve choice prediction. [8] showed the full DDM predicts choice probabilities better than the logit model. [9] proved that response times could enhance the identifiability of human preferences, compared to choices alone.
>
> Another application of response times is enhancing AI agents' decision-making. Dueling bandits and preference-based RL [14] typically use human choice models for preference elicitation. One popular choice model, the random utility model, can be derived from certain BAMs [9]. For example, both the Bradley-Terry model and EZ-DDM yield logistic choice probabilities (Eq.1 in our paper). To the best of our knowledge, **our work** is the first to leverage this connection to integrate BAMs within the framework of bandits (and RL). Note that our work lets the AI agent use RL to make decisions, which is different from [13] which models the human as an RL agent.
>
> - [[1] Ratcliff and McKoon 2008](https://ieeexplore.ieee.org/abstract/document/6796810)
> - [[2] Webb 2019](https://pubsonline.informs.org/doi/abs/10.1287/mnsc.2017.2931)
> - [[3] Wagenmakers et al. 2007](https://link.springer.com/article/10.3758/bf03194023)
> - [[4] Usher and McClelland 2001](https://psycnet.apa.org/record/2001-07628-003)
> - [[5] Fudenberg et al. 2018](https://www.aeaweb.org/articles?id=10.1257/aer.20150742)
> - [[6] Ratcliff et al. 2016](https://www.cell.com/trends/cognitive-sciences/abstract/S1364-6613(16)00025-5)
> - [[7] Krajbich 2019](https://www.sciencedirect.com/science/article/abs/pii/S2352250X18301866)
> - [[8] Clithero 2018](https://www.sciencedirect.com/science/article/abs/pii/S0167268118300398)
> - [[9] Alós-Ferrer et al. 2019](https://www.journals.uchicago.edu/doi/full/10.1086/713732)
> - [[10] Clithero 2018.](https://www.sciencedirect.com/science/article/abs/pii/S0167487016306444)
> - [[13] Pedersen et al. 2017](https://link.springer.com/article/10.3758/s13423-016-1199-y)
> - [[14] Bengs et al. 2021](https://www.jmlr.org/papers/volume22/18-546/18-546.pdf)
>
> ## Question: about other link functions that models human choices
> First, the EZ-DDM model's marginal distribution for choices (Eq.1 in our draft) coincides with the Bradley-Terry model. Therefore, we have adopted the logistic link function to form a fair comparison.
>
> Second, let's explore beyond the logistic link function. Suppose that the choice probability $\mathbb{P}[z_1\succ z_2]=\sigma(u_{z_1},u_{z_2})$, where $\sigma$ is a link function depending on the utilities $u_{z_1}$ and $u_{z_2}$. If we fix $u_{z_2}$ and only vary $u_{z_1}$, the function $\sigma(\cdot,u_{z_2})$ is known as a psychometric function, typically "S" shaped (see Fig.1.1 in [2]). This "S" shape means that as the human's preference strength becomes very large or very small, $\sigma(\cdot,u_{z_2})$ becomes flat and less informative, as you mentioned. In these circumstances, response times can be very helpful.
>
> Lastly, if we further assume that $\sigma$ depends only on the utility difference, $u_{z_1}-u_{z_2}$, this $\sigma$ becomes the link function commonly adopted in the preference learning literature. According to Sec.3.2 of [1], the usual assumptions are that $\sigma$ is strictly monotone in $(u_{z_1}-u_{z_2})$ and bounded within $[0, 1]$. Thus, as the utility difference becomes very large or very small, $\sigma(u_{z_1}-u_{z_2})$ becomes flat, so the same intuition holds.
>
> - [[1] Bengs et al. 2021](https://www.jmlr.org/papers/volume22/18-546/18-546.pdf)
> - [[2] Stochastic Choice Theory, Econometric Society Monograph](https://scholar.harvard.edu/sites/scholar.harvard.edu/files/tomasz/files/manuscript_01.pdf)

---

> > ### Comment · Reviewer_f1Vk · 2024-08-12
> >
> > Thank you for your detailed responses. I have updated my confidence score to 3 and intend to maintain my rating.

---

### Author Rebuttal · Authors · 2024-08-07

We address the two major concerns raised by multiple reviewers: the limited use of real-world datasets and the lack of non-asymptotic results.

## 1. New Simulation on a Real-world dataset
We present new simulation results based on another real-world response time dataset. This dataset [1] contains human binary choices and response times. In each query, each arm consists of two food items, and the human has an equal chance of obtaining either item after choosing that arm. For each user, we construct a bandit instance where the feature vector for each arm is composed of the user's ratings of the food items, augmented via second-order polynomials. For each user, an EZ-DDM is identified via Bayesian MCMC and then used as a simulator to generate human feedback. We compare the best-arm-identification errors over $100$ repetitions for three algorithms:
1. Transductive design with our choice-and-response-time estimator, denoted by $\left(\lambda\_{trans},\widehat{\theta}\_{CH,DT}\right)$.
2. Transductive design with a choice-only estimator, denoted by $\left(\lambda\_{trans},\widehat{\theta}\_{CH}\right)$.
3. Hard-query design with a choice-only estimator, denoted by $\left(\lambda\_{hard},\widehat{\theta}\_{CH}\right)$.

The results are plotted in Fig. 1 of our rebuttal PDF document. As shown, under various budgets and participant indices, incorporating response times (method (1), plotted in red) outperformed the other two methods.


## 2. New Non-asymptotic Analysis
We added lemmas stating non-asymptotic error probabilities on estimating the utility difference $x^{\top}\theta^*/a$ using both methods, i.e. combining response times with choices and using choices only. The results convey the same intuition as in the Thm. 3.1 and 3.2 in our draft:

*Response times make easy queries more useful. In other words, combining response times and choices, information from easy queries can be extracted more efficiently. This is not possible using choices only.*

We analyzed the non-asymptotic concentration for the utility difference estimated as the ratio between the empirical mean of choices and the empirical mean of response times, which appears on the right-hand side of Eq. 4 in our paper draft.
The error probability for such choice-and-response-time estimator is as follows:

**Lemma 1.** Consider any query $x\in\mathcal{X}$. For any scalar $\epsilon_r$ satisfying
\begin{equation}\begin{split}
    \epsilon_r \leq \min\left\\{\frac{x^{\top}\theta^*}{\sqrt{2}a}, \frac{(1+\sqrt{2})ax^{\top}\theta^*}{\mathbb{E}[t_x]}\right\\}
\end{split}\end{equation}
we have that
\begin{equation}\begin{split}
    \mathbb{P}\left(\left|\frac{\sum_{i\in[n_x]} c_{x,i}}{\sum_{i\in[n_x]} t_{x,i}} - \frac{x^{\top}\theta^*}{a}\right| > \epsilon_r\right)\leq 4\exp\left(-\frac{\left(\mathbb{E}[t_x]/(\sqrt{2}+2)\right)^2}{2} n_x\epsilon_r^2\right).
\end{split}\end{equation}


Alternatively, utility difference (or DDM drift) can be estimated using choices only (Eq. 5 in [1]). Converting $\mathbb{E}[c_x]\in[-1,1]$ to $\mathbb{E}[(c_x+1)/2]\in[0,1]$ and applying the logit function $h^{-1}(p)\colon=\text{logit}(p)=\log\left(p/(1-p)\right)$ estimates $2ax^{\top}\theta^*$.
The error probability for such choice-only estimator is as follows:

**Lemma 2.** Consider any query $x\in\mathcal{X}$. We have that
\begin{equation}\begin{split}
    \mathbb{P}\left(\frac{h^{-1}\left(\mathbb{E}[(c_x+1)/2]\right)}{2a^2}-\frac{x^{\top}\theta^*}{a} > \epsilon_r\right) \leq \exp\left(-\frac{\left(4a^2h'(2ax^{\top}\theta^*)\right)^2}{2}n_x\epsilon_r^2\right).
\end{split}\end{equation}
Here $h(x) = 1/\left(1+\exp(-x)\right)$.

For easy queries with $x^{\top}\theta^* \gg 1$, the factor $4a^2h'(2ax^\top\theta^*)$ in Lemma 2 is significantly smaller than factor $\mathbb{E}[t_x]/(\sqrt{2}+2)$ in Lemma 1. As a result, with $x^{\top}\theta^*\gg 1$, error probability of our estimation with both response times and choices is much smaller than that of the choice-only estimation.

The aforementioned two factors are plotted as functions of the utility difference $x^{\top}\theta^*$ in Fig. 2 of our rebuttal PDF document. Recall that this plot of non-asymptotic results looks similar to Fig. 2 (asymptotic results) in our paper draft. Indeed, they convey similar insights. In particular, When the human conservative parameter $a$ is small, for hard queries, the gray curve is slightly higher than the orange one, indicating that only using choices is slightly better. When $a$ is large, the gray curve is higher only for hard queries, while lower for easy queries. This conveys a similar insight as our Thm. 3.1 and 3.2, that using choice is better for hard queries, while using response time makes easy queries more useful.


- [[1] Wagenmakers et al. 2007](https://link.springer.com/article/10.3758/bf03194023)

---

### Decision · Program_Chairs · 2024-09-25

**Decision:**

Accept (oral)

**Comment:**

The paper studies preference-based linear bandits in fixed-budget best-arm identification problem. The authors investigate the benefit of incorporating human response time which inversely correlates with preference strength. The authors propose to incorporate the drift-diffusion model (DDM) from psychology with an elimination-based solution.  The paper presents both theoretical and empirical results showing that incorporating response time makes easy queries more useful and thus improves overall performance.  The reviewers are unanimously positive, recognizing the motivation and novelty of introducing response time and appreciating the solid analysis and clear paper presentation. The reviewers' questions about real-world datasets and the non-asymptotic results are addressed by the rebuttal. I recommend acceptance of the paper. The authors are suggested incorporate the discussion and new results from the rebuttal into the final version.